# Neural Networks beyond explainability:
# Selective inference for sequence motifs

**Antoine Villié**                                        *antoine.villie@univ-lyon1.fr*
*Université de Lyon, Université Lyon 1, CNRS, VetAgro Sup, Laboratoire de Biométrie et Biologie Evolutive, UMR5558, Villeurbanne, France*

**Philippe Veber**                                        *philippe.veber@univ-lyon1.fr*
*Université de Lyon, Université Lyon 1, CNRS, VetAgro Sup, Laboratoire de Biométrie et Biologie Evolutive, UMR5558, Villeurbanne, France*

**Yohann De Castro**                                        *yohann.de-castro@ec-lyon.fr*
*Institut Camille Jordan, École Centrale Lyon, CNRS UMR 5208*
*Institut universitaire de France (IUF)*

**Laurent Jacob**                                        *laurent.jacob@cnrs.fr*
*Sorbonne Université, CNRS, IBPS, Laboratory of Computational and Quantitative Biology (LCQB), UMR 7238, Paris 75005, France*

**Reviewed on OpenReview:** *https://openreview.net/forum?id=nddEHTSnqg*

## Abstract

Over the past decade, neural networks have been successful at making predictions from biological sequences. As in other fields of deep learning, tools have been devised to extract features such as sequence motifs that can explain the predictions made by a trained network. Here we intend to go beyond explainable machine learning and introduce SEISM, a selective inference procedure to test the association between these extracted features and the predicted phenotype. In particular, we discuss how training a one-layer convolutional network is formally equivalent to selecting motifs maximizing some association score. We adapt existing sampling-based selective inference procedures by quantizing this selection over an infinite set to a large but finite grid. Finally, we show that sampling under a specific choice of parameters is sufficient to characterize the composite null hypothesis typically used for selective inference—a result that goes well beyond our particular framework. We illustrate the behavior of our method in terms of calibration, power and speed and discuss its power/speed trade-off with a simpler data-split strategy. SEISM paves the way to an easier analysis of neural networks used in regulatory genomics, and to more powerful methods for genome wide association studies (GWAS).

## 1 Introduction

In the recent years, neural networks have been successfully used for making predictions from biological sequences. In particular, they have brought significant improvements in regulatory genomics, *e.g.* to predict cell-type specific transcription factor binding, gene expression, chromatin accessibility or histone modifications from a DNA sequence (Zhou & Troyanskaya, 2015; Kelley et al., 2018; Avsec et al., 2021a;b). These tasks are expected to be a good proxy for predicting the functional effect of non-coding variants, and help us in turn make better sense of the observed human genetic variation and its effect on various phenotypical traits including diseases. Most successful models have used convolutional neural networks (CNNs, LeCun & Bengio, 1998) and more recent approaches have explored self-attention mechanisms (Vaswani et al., 2017). These models have been trained from experimental data obtained from ChIP-seq, ATAC-seq, DNase-seq, or CAGE assays, that provide examples where both the DNA sequence and the outcome of interest are known.

A commonly outlined limitation of neural networks is their lack of explainability or black box aspect, *i.e.*, the contrast between their excellent prediction accuracy and the possibility to explain these in intuitive or mechanistic terms (Ras et al., 2022; Molnar, 2022). Elementary one-layer CNNs don't face this issue, as their trained filters have a straightforward interpretation as position weight matrices (PWMs, Harr et al., 1983; Schneider & Stephens, 1990), a historical and basic element of regulatory genomics. Nonetheless, these simple models are notoriously too simple to capture the complexity of the regulatory code which requires to account not only for individual motif presence but for their long range sequence context and mutual interactions (Avsec et al., 2021b). Multi-layer CNNs and self-attention mechanisms model this additional complexity but are less straightforward to interpret. Tools inspired from the explainable deep learning literature have been adapted to extract features beyond PWMs and one-layer CNNs to explain the predicted regulatory behavior (Novakovsky et al., 2022). It is therefore often possible to explain the predictions of a trained neural network for biological sequences, either directly through estimates of its parameters or through features extracted post hoc.

Unfortunately, finding features somewhat associated to an outcome is often not enough, as an observed non-zero association can be spurious. In experimental science, it is actually common to quantify the significance of this association, *e.g.*, by testing the hypothesis that it is zero. Genome wide association studies (GWAS, Visscher et al., 2017) for example find genetic variants correlated with a trait by building a linear model explaining this trait by each variant and testing the hypothesis that the weight is zero. Statistical significance has its own limitations (Wasserstein & Lazar, 2016), but often provides an intuitive scale for identifying relevant features. Quantifying the significance of associations between interpretable features and predicted outcome is equally important in the context of neural networks but has received little attention to our knowledge.

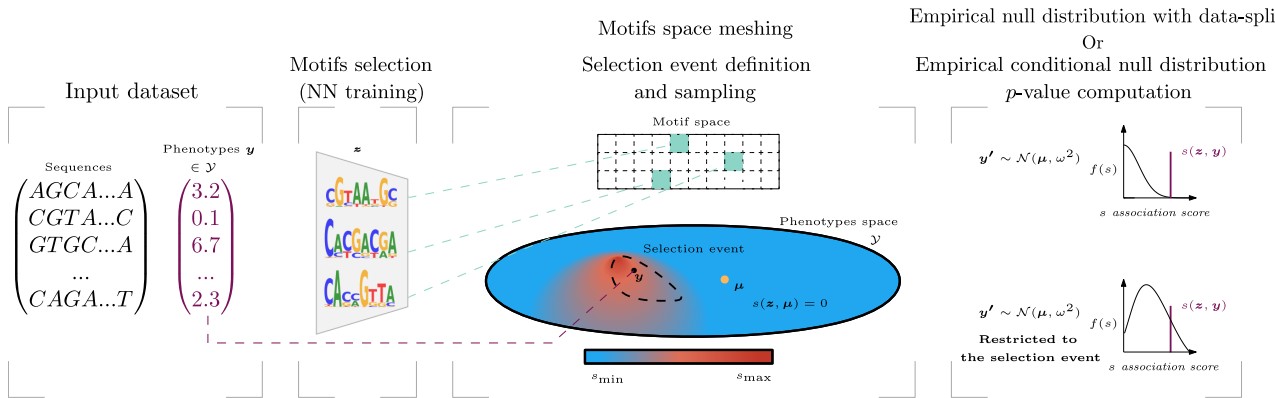

Figure 1: Overview of our SEISM procedure. (a) The input is a set of sequences and corresponding phenotypes in some space $\mathcal{Y}$ (b) It trains a convolutional neural networks to predict a phenotype from sequences, which leads to the selection of sequence motifs. (c) Then SEISM partitions the space of motifs to quantize the selection. The selection event is the set of phenotype vectors that would lead to selecting an element in the same mesh. (d) Using a sampling strategy, SEISM builds a null distribution for the test statistic, conditional to the selection event. The $p$-values associated with a selected motif is the quantile of its score under this distribution.

Here, we set out to go beyond explainable machine learning by introducing SElective Inference for Sequence Motifs (SEISM), depicted in Figure 1, a valid statistical inference procedure for these features. In order to do so, we cast commonly used CNNs in a feature selection framework, and show that it achieves similar selection performances as existing bioinformatics algorithms on *de novo* motifs discovery tasks. This selection needs to be accounted for when testing the association of the features with the predicted trait. This problem has been discussed and addressed in the growing literature on post-selection inference over the past few years, using *e.g.* data-split (Wasserman & Roeder, 2009) or selective inference strategies (Taylor & Tibshirani, 2015; Reid et al., 2018; Slim et al., 2019). The former split the data into two parts, performing selection on one and inference on the other. They produce valid inference but necessarily result in a reduction of the sample

size, which is unsatisfying when the original sample size is limited. By contrast, the latter condition the null distribution on the selecting event, which generally provides more power but can prove more computationally intensive.

Our contributions are as follows:

- We formally cast one-layer CNNs into a motif discovery tool, reaching similar performances as *de-novo* motifs discovery tools from the bio-informatics literature (Section 3).

- We define a post-selection inference framework for the features selected by the neural network, using either data-split or selective inference (Section 4), each being more appropriate in a given sample size regime.

- Both strategies require sampling under a normal null hypothesis which is composite—several mean vectors define the same null—and depends on an unknown parameters. We provide invariance results suggesting a practical procedure that works around these issues (Section 4.6). To our knowledge, they were a blind spot in sampling-based post-selection inference approaches beyond our specific context.

- Existing selective inference methods are only defined for selections over a finite set. We work around this issue by quantizing our selection to a very large but finite space, making it amenable to existing sampling strategies. We show that the resulting procedure is well calibrated.

- We provide a PyTorch implementation of SEISM at:
  https://gitlab.in2p3.fr/antoine.villie1/seism.

In this paper, we restrict our presentation to simple one-layer CNNs and sequence motifs. The procedure we introduce here, however, is not limited to this framework. It can be applied to more expressive features proposed in the explainable machine learning literature, but may require some further work depending on the feature considered.

## 2 A short overview of our SEISM procedure

SEISM aims to detect sequence motifs associated with a biological outcome, and to test the statistical significance of this association. Here we briefly describe the selective inference version of SEISM in order to give the reader an overview of the procedure. It is summarized in Algorithm 1, and more details will be given in the following sections.

(i) SEISM takes as input biological sequences $\boldsymbol{X}$ associated with a phenotype $\boldsymbol{y}$. The user must also specify the number of motifs to find, as well as a parameter controlling the meshing of the motif space, that is the precision with which the found motifs will be tested.

(ii) The motif selection step corresponds to the maximisation of a so-called association score $s(\cdot, \cdot)$, which depends on the phenotype and on the motifs $\boldsymbol{z}$ through their activation patterns in the biological sequences $\boldsymbol{\varphi}^{\boldsymbol{z}, \boldsymbol{X}}$. This step is formally equivalent to training a one hidden layer CNN. We implement a greedy procedure, optimizing each new filter over the residuals of the previously entered ones, using a gradient descent method initialized at the $k$-mer with the best score. To that end, we enumerates the $k$-mers contained in $\boldsymbol{X}$ using the DSK software (Rizk et al., 2013) and compute their scores $s(\cdot, \cdot)$.

(iii) SEISM splits the set of sequence motifs into meshes according to the input parameter. This step leads to the definition of a set of null hypotheses and of a selection event $E$, *i.e.* the set of outcomes $\boldsymbol{y}'$ that would have led to the selection of motifs within the same meshes as the ones selected in (ii), namely the sequence of meshes $(M_{i_1}, \ldots, M_{i_q})$. Formally, the selection event reads

$$E := \left\{ \boldsymbol{y}' \in \mathcal{Y} \, : \, \forall j \in [q], \; \arg\max_{\boldsymbol{z} \in \mathcal{Z}} s(\boldsymbol{z}, \boldsymbol{P}_j \boldsymbol{y}') \in M_{i_j} \right\}, \tag{1}$$

for some projection matrix $\boldsymbol{P}_j$, to be defined later.

(iv) It approximates the conditional null distribution of the test statistics by sampling biological outcomes $\boldsymbol{y'}$ under the null, conditionally to the selection event. This sampling is performed using a hit-and-run strategy (according to Algorithm 2), by building a discrete time Markov chain on $E$ whose distribution converges to the uniform one.

(v) SEISM finally computes the $p$-values for the null hypotheses defined in (iii), associated with the selected motifs in ii, using the empirical distribution of the test statistics, and returns the motifs with their association $p$-values. Given these $p$-values, one can adjust the number of selected motifs discarding the ones with non-significant $p$-values. This multiple-testing issue has not been investigated in this paper, but the practitioner can use for instance a Bonferroni bound to select the number of motifs.

The data-split version of SEISM applies the same (i)-(ii) steps on a fraction of the data , and simply compares the scores of the selected motifs *on the remaining data* to the distribution of scores for the same motif with data sampled under the null distribution—as opposed to the selective null generated by (iii)-(v). Sampling is much faster under the null than under the selective null, because it does not involve a rejection step. Both samplings will need our results in Section 4.6 to avoid depending on a particular value of the mean and variance parameters.

---

**Algorithm 1** SEISM algorithm (general formulation)

---

# **Description:** SEISM *selects a set of sequence motifs* $(\boldsymbol{z}_1, \ldots, \boldsymbol{z}_q)$ *based on an association score* $s(\cdot, \cdot)$, *and evaluate their p-values based on a partition* $\mathcal{Z} = \bigsqcup M_i$.

**Inputs:** *Response* $\boldsymbol{y} \in \mathcal{Y} \subseteq \mathbb{R}^n$, *sequence samples* $\boldsymbol{X}$, *feature function* $\boldsymbol{z} \in \mathcal{Z} \mapsto \boldsymbol{\varphi^{z,X}} \in \mathbb{R}^n$, *association score* $s : \mathcal{Z} \times \mathcal{Y} \to \mathbb{R}$, *number of selected motifs* $q \geq 1$, *meshes* $\mathcal{Z} = \bigsqcup\limits_{i=1} M_i$, *sampling algorithm* $\mathcal{HR}$.

**Result:** $((p_1, \boldsymbol{z}_1), \ldots, (p_q, \boldsymbol{z}_q))$, sequence of $p$-values and sequence motifs.

# **Selection step:** *Selection of the sequence motifs* $(\boldsymbol{z}_1, \ldots, \boldsymbol{z}_q)$ *and the sequence of meshes* $(M_{i_1}, \ldots, M_{i_q})$.

1 **for** $j = 1, \ldots, q$ **do**
2     $\boldsymbol{z}_j \leftarrow \underset{\boldsymbol{z} \in \mathcal{Z}}{\arg\max}\, s(\boldsymbol{z}, \boldsymbol{P}_j \boldsymbol{y})$ ;           `//` $\boldsymbol{P}_k$ `orthogonal projection onto` $\underset{\ell < j}{\mathrm{Span}}\left\{\boldsymbol{\varphi^{z_\ell, X}}\right\}^{\perp}$
3     $i_j \leftarrow i$ s.t. $\boldsymbol{z}_j \in M_i$ ;                  `// the mesh` $M_{i_j}$ `is selected`
4 **end**

# **Inference step:** SEISM *provides a p-value* $p_k$ *on the statistical influence of the selected sequence motifs* $\boldsymbol{z}_k$ *conditional on the selection event* (1) *of observations* $\boldsymbol{y'}$ *that would have led to same selection of the sequence of meshes* $(M_{i_1}, \ldots, M_{i_q})$.

5 $\boldsymbol{y'}^{(1)}, \ldots, \boldsymbol{y'}^{(N)} \leftarrow \mathcal{HR}(y, (M_{i_1}, \ldots, M_{i_q}))$ ;     `// Sampling outcomes under the selected null`

6 **for** $j = 1, \ldots, q$ **do**
7     $\tilde{F}_j(\cdot\,; \boldsymbol{y'}^{(1)}, \ldots, \boldsymbol{y'}^{(N)}) \leftarrow$ empirical cumulative distribution function of $s(\boldsymbol{r}M_{i_j}, \boldsymbol{\Pi_j} \boldsymbol{y'})$ under the selected null ;     `//` $\boldsymbol{r}M_{i_j}$ `is a motif representing` $M_{i_j}$ `and` $\boldsymbol{\Pi_j}$ `the orthogonal projection onto` $\underset{\ell \neq j}{\mathrm{Span}}\left\{\boldsymbol{\varphi^{z_\ell, X}}\right\}^{\perp}$
8     $p_j \leftarrow \tilde{F}_j(s(\boldsymbol{r}M_{i_j}); \boldsymbol{y'}^{(1)}, \ldots, \boldsymbol{y'}^{(N)})$ ;              `// output the` $j^{\text{th}}$ `p-value`
9 **end**

---

# 3    One hidden layer CNNs select sequence motifs maximizing an association score

One-layer CNNs have been at the core of the rising popularity of deep learning over the past decade, by enabling major improvements in computer vision tasks (Krizhevsky et al., 2012). Although they are formally a specialized fully connected feedforward networks with additional constraints on the weights, CNNs are equivalent to, and more often thought of as, a set of *convolutions* of the vectorial input with some smaller vectors referred to as filters. When applying the network, dot products are taken between each of them and

successive windows of the vectorial input followed by some non-linear operation, producing an activation profile for each filter. In one-layer networks, these activations are pooled across the windows into a single scalar for each filter and these scalars are combined—typically through a linear or regular fully connected network—to provide a prediction for the input. Because convolution filters are homogeneous to the input, they easily lend themselves to interpretation: as small image patches for image inputs, and as sequence motifs for appropriately encoded biological sequence inputs. Accordingly, activation profiles reflect how much each piece of the input is similar to the filter—in the sense of the dot product—and applying a one-layer CNN amounts to applying a predictive function to a modified representation of the original data by these similarity profiles. Because convolution filters are jointly optimized with the parameterization of the predictive function, CNNs are often described as a strategy to jointly learn a data representation and a function acting on this representation, both being optimized for a prediction objective. In computer vision, the optimized filters of the first layer typically learn to detect edges with different orientations. In biological sequences, they learn short sequences whose presence anywhere in the input is predictive of the output phenotype used for training.

$$
\begin{array}{c}
A \\
C \\
G \\
T
\end{array}
\begin{pmatrix}
0.11 & 0.27 & 0.01 & 0.99 & 0.67 & 0.77 & 0.00 & 0.00 \\
0.00 & 0.48 & 0.00 & 0.00 & 0.25 & 0.01 & 0.60 & 0.00 \\
0.01 & 0.17 & 0.13 & 0.00 & 0.05 & 0.21 & 0.40 & 0.00 \\
0.88 & 0.08 & 0.86 & 0.01 & 0.03 & 0.01 & 0.00 & 1.00
\end{pmatrix}
$$

Figure 2: A motif represented by its position weight matrix and corresponding sequence logo. The total height of the letters indicates the information content of the position (in bits), closely related to the Shannon entropy.

Unlike input sequences that are formed by a discrete succession of letters in some alphabet, trained filters are continuous and therefore account for possible variation in the predictive short sequence, *e.g.*, a T mostly followed by a C but sometimes an A or a G and so on (Figure 2). These probabilistic objects have also been used for a long time in the bioinformatics literature and referred to as position weight matrices (PWMs). Inferring PWMs either according to their frequency in a set of sequences (Bailey et al., 2006) or their discriminating power between two sets (Bailey, 2021) has been a major theme over the past thirty years. Here we formalize the training a one-layer CNN as equivalent to the selection of a set of sequence motifs that are optimal for some association score. This formalization will be instrumental in the definition of our hypothesis testing procedure in Section 4.

**Notations**  Let $\boldsymbol{X}$ represent a data set of $n$ one-hot encoded sequence samples $\{\boldsymbol{x}_1, \boldsymbol{x}_2, \ldots, \boldsymbol{x}_n\}$, in a set $\mathcal{X}$ of biological sequences assumed to be over an alphabet $\mathcal{A}$—for DNA sequences, $\mathcal{A} = \{A, C, T, G\}$. One-hot encoding maps each letter in $\mathcal{A}$ to a vector in $\{0, 1\}^{|\mathcal{A}|}$, with all-zero entries except for a single 1 at the coordinate corresponding to the order of the letter in $\mathcal{A}$—for DNA sequences, $A$ is encoded as $(1, 0, 0, 0)$. Every $\boldsymbol{x}_i$ is therefore encoded as a matrix in $\{0, 1\}^{|\mathcal{A}| \times |\boldsymbol{x}_i|}$—although in practice, encoded sequences are often padded with dummy columns to have the same lengths. We denote $y_i \in \mathcal{Y}$ the measurement of a biological property associated with sequence $\boldsymbol{x}_i$, and $\boldsymbol{y} \in \mathcal{Y}^n$ the corresponding vector of outcomes. We consider one-layer CNNs with a Gaussian non-linearity with scale $\omega$, a max global pooling and a linear prediction function. These CNNs parameterize a function $f : \mathcal{X} \to \mathcal{Y}$ by $q$ filters of length $k$, namely $\boldsymbol{Z} := \{\boldsymbol{z}_1, \ldots, \boldsymbol{z}_q\} \in \mathcal{Z}^q$, where $\mathcal{Z}$ is a subset of $\mathbb{R}^{|\mathcal{A}| \times k}$, given by the simplex in this paper:

$$
\mathcal{Z} = \left\{ \boldsymbol{z} \in \mathbb{R}_+^{|\mathcal{A}| \times k} \; : \; \forall j \in [k], \; \sum_{i=1}^{|\mathcal{A}|} z_{i,j} = 1 \right\}, \tag{2}
$$

and $q$ weights $\beta \in \mathbb{R}^q$.

More precisely, we define $f(\boldsymbol{x}_i) := \left(\boldsymbol{\varphi}^{\boldsymbol{Z},\boldsymbol{X}}\boldsymbol{\beta}\right)_i$, with $\boldsymbol{\varphi}^{\boldsymbol{Z},\boldsymbol{X}} \in \mathbb{R}^{n \times q}$ defined as $\boldsymbol{\varphi}^{\boldsymbol{Z},\boldsymbol{X}} = \boldsymbol{C}_n\tilde{\boldsymbol{\varphi}}^{\boldsymbol{Z},\boldsymbol{X}}$, where $\boldsymbol{C}_n = I_n - n^{-1}\mathbf{1}_n\mathbf{1}_n^\top$ is the centering operator, $I_n$ the identity matrix, $\mathbf{1}_n$ the all-one vector in $\mathbb{R}^n$, and

$$\tilde{\varphi}_{i,j}^{\boldsymbol{Z},\boldsymbol{X}} := \max_{\boldsymbol{u} \in [\boldsymbol{x}_i]_\ell} \left\{ \exp\left(-\frac{||\boldsymbol{z}_j - \boldsymbol{u}||^2}{2\omega^2}\right) \right\}, \tag{3}$$

where $[\boldsymbol{x}_i]_\ell$ denotes the set of $\ell$ consecutive entries of the vector $\boldsymbol{x}_i$ (and of its reverse-complement counterpart), and $\omega$ is a bandwidth hyperparameter whose impact and tuning is studied in Appendix A. This model differs with a typical CNN in two ways. First, it uses a Gaussian activation function instead of an exponential one; second the use of the centering operator that sets the average of the activation to zero. These adjustments were made to improve the SEISM algorithm's selection performances.

## 3.1 From empirical risk minimization to association scores

The function $f$ is learned in a classical penalized empirical risk minimization framework, using the data $\{\boldsymbol{X}, \boldsymbol{y}\}$:

$$\min_{(\boldsymbol{Z},\boldsymbol{\beta}) \in (\mathcal{Z} \times \mathbb{R}^q)} n^{-1}\left\|\boldsymbol{y} - \boldsymbol{\varphi}^{\boldsymbol{Z},\boldsymbol{X}}\boldsymbol{\beta}\right\|^2 + \lambda\|\boldsymbol{\beta}\|^2, \tag{4}$$

for some $\lambda > 0$. Equation (4) formalizes the idea that learning a one-layer CNN on one-hot encoded sequences amounts to learning a data-representation $\boldsymbol{\varphi}^{\boldsymbol{Z},\boldsymbol{X}}$ of the sequences parameterized by a set $\boldsymbol{Z}$ of filters—corresponding to PWMs—and a linear function with weights $\boldsymbol{\beta}$ acting on this representation. Noting that there exists a unique explicit optimal $\boldsymbol{\beta}$ for Eq. (4), it follows immediately that:

$$\arg\min_{\boldsymbol{Z}} \left\{ \min_{\boldsymbol{\beta}} \left\{ n^{-1}\|\boldsymbol{y} - \boldsymbol{\varphi}^{\boldsymbol{Z},\boldsymbol{X}}\boldsymbol{\beta}\|^2 + \lambda\|\boldsymbol{\beta}\|^2 \right\} \right\} = \arg\max_{\boldsymbol{Z}} \left\{ s_\lambda^{\text{ridge}}(\boldsymbol{Z}, \boldsymbol{y}) \right\}, \tag{5}$$

where $s^{\text{ridge}}$ defines a particular quadratic association score between an outcome $\boldsymbol{y}$ and a set of filters $\boldsymbol{Z}$:

$$s_\lambda^{\text{ridge}}(\boldsymbol{Z}, \boldsymbol{y}) := \boldsymbol{y}^T\boldsymbol{\varphi}^{\boldsymbol{Z},\boldsymbol{X}}\left[(\boldsymbol{\varphi}^{\boldsymbol{Z},\boldsymbol{X}})^T\boldsymbol{\varphi}^{\boldsymbol{Z},\boldsymbol{X}} + \lambda n\boldsymbol{I}_q\right]^{-1}(\boldsymbol{\varphi}^{\boldsymbol{Z},\boldsymbol{X}})^T\boldsymbol{y}. \tag{6}$$

It formalizes the training of a CNN as the selection of a set of filters whose association with $\boldsymbol{y}$ in the sense of $s_\lambda^{\text{ridge}}$ is maximal. Of note, one has

$$\lim_{\lambda \to \infty} \lambda n \times s_\lambda^{\text{ridge}}(\boldsymbol{Z}, \boldsymbol{y}) = \boldsymbol{y}^T\boldsymbol{\varphi}^{\boldsymbol{Z},\boldsymbol{X}}(\boldsymbol{\varphi}^{\boldsymbol{Z},\boldsymbol{X}})^T\boldsymbol{y} =: s^{\text{HSIC}}(\boldsymbol{Z}, \boldsymbol{y}),$$

so for large values of the regularization hyperparameter, selecting filters by learning a CNN is equivalent to selecting filters with the classical HSIC score (Song et al., 2012), because $\boldsymbol{\varphi}$ already includes a centering operator. In addition to connecting $s^{\text{ridge}}$ with $s^{\text{HSIC}}$, we observed that the centering in the definition of $\boldsymbol{\varphi}^{\boldsymbol{Z},\boldsymbol{X}}$ led to the selection of better sequence motifs in our experiments. Observe that the centering matrix is an orthogonal projection matrix onto $\mathcal{E} := \text{Range}(\boldsymbol{C}_n)$, the orthogonal of the vector line generated by the vector $\mathbf{1}$, and it holds

$$\left\|\boldsymbol{y} - \boldsymbol{\varphi}^{\boldsymbol{Z},\boldsymbol{X}}\boldsymbol{\beta}\right\|_n^2 = \left\|\boldsymbol{C}_n\boldsymbol{y} - \boldsymbol{\varphi}^{\boldsymbol{Z},\boldsymbol{X}}\boldsymbol{\beta}\right\|_n^2 + \left\|\boldsymbol{y} - \boldsymbol{C}_n\boldsymbol{y}\right\|_n^2. \tag{7}$$

The solution of (4) is unchanged if $\boldsymbol{y}$ is replaced by $\boldsymbol{C}_n\boldsymbol{y}$, and so we can assume that $\boldsymbol{y} \in \mathcal{E}$ without any generality loss. Furthermore, this shows that we can work with skewed data in a classification context, since imbalanced classes will have no effect on the result.

## 3.2 Greedy optimization

It is common to solve (4) by stochastic gradient descent (SGD) jointly over the $q$ filters. More generally, this approach for training a neural network with a single, large hidden layer is known to find a global optimizer at the large $q$ limit under some assumptions (Soltanolkotabi et al., 2019). Our objective here is slightly different: we do not necessarily aim at approximating a continuous measure with a large number of particles, but we aim at selecting a small number of particles lending themselves to a biological interpretation. Furthermore, the number of relevant motifs on a given dataset is generally unknown. In this context, it is known that jointly optimizing the convolution filters leads to irrelevant PWMs, with some actual motif split across several filters

and other duplicated (Koo & Eddy, 2019). A possible strategy is to forego filter-level interpretation, train an overparameterized network—with a much larger $q$ than the expected number of motifs—and use attribution methods to extract relevant motifs or other interpretable features from the trained network (Shrikumar et al., 2018). Here we adopt a different strategy using a forward stepwise procedure, where we iteratively optimize each of the convolution filters over the residual error left by the previous ones.

More precisely at each of the $q$ steps, we select $\boldsymbol{z}_j$ such that:

$$\boldsymbol{z}_j = \arg\max_{\boldsymbol{z} \in \mathcal{Z}} s^{\text{ridge}}(\boldsymbol{z}, \boldsymbol{P}_j \boldsymbol{y}), \tag{8}$$

where $\boldsymbol{P}_j$ is the projection operator onto the orthogonal of the subspace $\text{Span}_{\ell < j}\left\{\mathbf{1}, \boldsymbol{\varphi}^{\boldsymbol{z}_\ell, \boldsymbol{X}}\right\}$, see line 2 of Algorithm 1. This is how $\boldsymbol{z}_j$ is optimized over the residuals of the previous filters. The vector $\mathbf{1}$ enforces that we project on a subspace of $\mathcal{E}$, in particular $\boldsymbol{P}_1 = \boldsymbol{C}_n$. Without this projection, iterating (8) would return the same $\boldsymbol{z}$. Of note, joint optimization procedures of the $q$ filters don't face this issue, and forward selection procedures over finite sets of features work around the problem by iteratively removing the selected elements from the set over which selection if performed (Slim et al., 2019). This sequential strategy combined with the testing procedure introduces in Section 4 provides a data-driven mean to choose the number $q$ of relevant motifs.

In practice, we solve (8) with a standard gradient descent algorithm, initialized at the $k$-mer with the best association score. The $k$-mer list is obtained using the DSK software (Rizk et al., 2013). The length $k$ first varies according to a user-defined range, and the optimal value is chosen by SEISM, as described in Appendix A. We work on a less constrained set than $\mathcal{Z}$ (2) and don't enforce the positivity constraint during optimization. We project the optimized motifs onto the full $\mathcal{Z}$ at the end of the process. Our procedure also requires to choose a motif length $k$. We proceed adaptively by choosing the length leading to the highest score, within a user-specified range.

With the one-layer CNNs training formally cast as the successive selection of $q$ sequence motifs optimizing an association score, we now turn to the problem of testing the significance of these associations. Of note, what follows is only based on the definition of an association score and could be applied to perform inference on other features coming from the training step of any algorithm, as long as one can define an association score between the feature and the outcome.

## 4 Post-selection testing of the association between the outcome and trained convolution filters

We now turn to the problem of testing the association between the selected motifs $\boldsymbol{z}$ and the trait $\boldsymbol{y}$. In order to do so, we need to solve three interrelated problems. First, the motifs were specifically selected for their association with the trait, which leads to the well known post-selection inference problem. Any inference procedure that disregards that the hypothesis was constructed using the same data used for testing is likely invalid and produces deflated $p$-values. Second, we deal with a continuous selection event, because (8) is performed over a continuous set $\mathcal{Z}$. By contrast, existing solutions for post-selection inference address selections over finite sets. Third, the null hypothesis commonly used for similar post-selection inference problems is composite, *i.e.*, it corresponds to several values of the parameters. Existing methods work around this issue by fixing theses parameters to arbitrary values, thereby limiting the scope under which they are calibrated. Here we present our solutions to these three problems.

Consider the Gaussian model:

$$\boldsymbol{y} = \boldsymbol{\mu} + \sigma\boldsymbol{\epsilon} \tag{9}$$

where $\boldsymbol{\mu} \in \mathcal{E}$ is the target deterministic signal, and $\boldsymbol{\epsilon} \sim \mathcal{N}(\mathbf{0}, \boldsymbol{C}_n)$ the standard Gaussian distribution on $\mathcal{E}$.

### 4.1 Selective null hypothesis

We follow Yamada et al. (2018) and test the association of a motif $\boldsymbol{z}$ through the following null hypothesis:

$$\mathbb{H}_0 : \text{``}s(\boldsymbol{z}, \boldsymbol{\mu}) = 0\text{''}, \tag{10}$$

for some association score $s$. For a $\boldsymbol{z}$ chosen independently of the data, $\mathbb{H}_0$ could be tested by sampling $\boldsymbol{y}'$ under the corresponding distribution, and using the quantile of the $s(\boldsymbol{z}, \boldsymbol{y}')$ scores corresponding to $s(\boldsymbol{z}, \boldsymbol{y})$ as a $p$-value—*i.e.*, the probability when sampling under $\mathbb{H}_0$ to observe a score as extreme as $s(\boldsymbol{z}, \boldsymbol{y})$. In our case, however, the motifs $\boldsymbol{z}$ in the trained convolution filters were specifically selected for their strong association with $\boldsymbol{y}$, and this procedure would not produce calibrated $p$-values. This problem is known as post-selection inference, and has been discussed and addressed in a growing literature over the past few years. Data-split strategies lead to valid inference but necessarily result in a reduction of the sample size, which is unsatisfying when the original sample size is limited. Alternatively, selective inference frameworks were developed in the recent years to address these issues. We refer to (Hastie et al., 2015, Chapter 6) and references therein for a general presentation. Taylor et al. (2014) and Lee et al. (2016) address scenarios where the selection event, *i.e.* the set of data outputs that would result in the selection of the same set of features, is polyhedral— determined by the finite intersection of linear constraints. Reid & Tibshirani (2013), and later Reid et al. (2015) extend this selection to clusters or groups of features, still in the linear framework. Yamada et al. (2018) extended post-selection inference to the non-linear framework, by proposing a kernel-based approach, where the selection is performed through the HSIC criterion. Slim et al. (2019) generalize this work, by allowing the selection to be carried out with a wider range of tools, making use of quadratic association scores.

To our knowledge, the selective inference literature only addresses the problem of selecting features from a discrete collection and does not provide a solution for selections from a continuous set like our $\mathcal{Z}$. Hence, testing (10) directly is not feasible and we resort to the quantization of the motif space to address this problem.

In addition to that, we push the analysis of the statistical model further, in order to be able to apply it with weaker assumptions on the data distribution.

## 4.2 Selective inference over a continuous set of features

Formally, our selection event $E_{\text{cont.}}$ is the set of outcomes $\boldsymbol{y}'$ that would have led to the selection of the same set of motifs $\boldsymbol{Z} = \{\boldsymbol{z}_1, \ldots, \boldsymbol{z}_q\}$ than the one selected using $\boldsymbol{y}$ from the real dataset, when applying the same selection procedure:

$$E_{\text{cont.}} := \left\{ \boldsymbol{y}' \in \mathcal{E} \ : \ \forall j \in \{1, \ldots, q\} \ \arg\max_{\boldsymbol{z} \in \mathcal{Z}} s(\boldsymbol{z}, \boldsymbol{P}_j \boldsymbol{y}') = \boldsymbol{z}_j \right\}, \tag{11}$$

where $\boldsymbol{P}_j$ is the orthogonal projection onto $\underset{\ell < j}{\text{Span}} \left\{ \mathbf{1}, \boldsymbol{\varphi}^{\boldsymbol{z}_\ell, \boldsymbol{X}} \right\}^\perp$.

A simple rejection approach to sample from the null (10) conditioned to $E_{\text{cont.}}$ would be to sample $\boldsymbol{y}$ in $\mathcal{E}$ under (9, 10) and only retain those in $E_{\text{cont.}}$. Unfortunately, $E_{\text{cont.}}$ belongs to a strictly lower-dimensional vector space of $\mathbb{R}^n$ and is therefore a null set for the Lebesgue measure on $\mathbb{R}^n$. For $s^{\text{HSIC}}$ and $s^{\text{ridge}}$, and noting that a maximum is also a critical point, we indeed obtain:

$$\boldsymbol{y}' \in E_{\text{cont.}} \implies \forall j \in \{1, \ldots, q\} \ \boldsymbol{P}_j \boldsymbol{y}' \in \text{Span} \left\{ \nabla_{\boldsymbol{z}} \boldsymbol{\varphi}^{\boldsymbol{z}_j, \boldsymbol{X}} \right\}^\perp.$$

For $q = 1$ and assuming that the different directions of the gradient are independent, this spans is a vector subspace with dimension $n - 4 \times k$. We empirically observed that sampling from this subspace produced a non-zero proportion of $\boldsymbol{y}'$ in $E_{\text{cont.}}$. Nonetheless, choosing a sampling distribution that leads to the correct conditional distribution is not straightforward—and may not even be possible—as discussed in Supplementary Material B. Moreover, relying on conditional probability with respect to a null set is not well defined and may lead to the Borel-Kolmogorov paradox (Bungert & Wacker, 2022), which further complicates its use.

We choose to circumvent this issue using a partition of the space $\mathcal{Z}$ of motifs spaces, over which our selection (8) operates, into a very large but finite set of meshes: $\mathcal{Z} = \bigsqcup M_i$. As depicted in Figure 3, we consider a regular partition of each coordinates into $m$ bins:

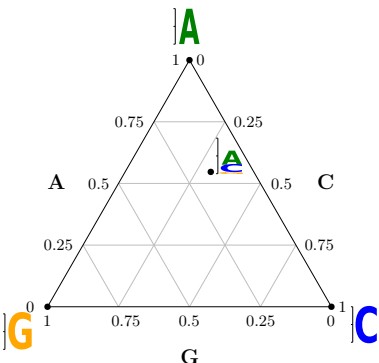

Figure 3: Discretization of the 3-letters alphabet simplex $\{A, C, G\}$, with a binning parameter for the meshes $m = 4$.

Based on this partition into meshes, we define a quantized selection event $E$ as follows. First, given an outcome $\boldsymbol{y}$ we define the sequence of the $q$ selected meshes $(M_{i_1}, \ldots, M_{i_q})$ as

$$\forall j \in \{1, \ldots, q\} \ , \ \arg\max_{\boldsymbol{z} \in \mathcal{Z}} s(\boldsymbol{z}, \boldsymbol{P}_j \boldsymbol{y}) \in M_{i_j} \ ,$$

Second, the selection event is given by:

$$E(i_1, \ldots, i_q) := \left\{ \boldsymbol{y}' \in \mathcal{Y} \ : \ \forall j \in \{1, \ldots, q\} \ , \ \arg\max_{\boldsymbol{z} \in \mathcal{Z}} s(\boldsymbol{z}, \boldsymbol{P}_j \boldsymbol{y}') \in M_{i_j} \right\}, \tag{12}$$

the set of outcomes $\boldsymbol{y}'$ that would have led to the selection of motifs within the same meshes as the selected ones $(M_{i_1}, \ldots, M_{i_q})$.

We now show how quantization (12) of the selection problem make enables the definition of a valid inference procedure. We start with the simplest case where we select a single motif ($q = 1$).

### 4.3 Test with only one motif $q = 1$, $\mu$ and $\sigma$ fixed

In this section, considering the motif $\boldsymbol{z}_1$ was chosen by the SEISM selection procedure, selection event (12) boils down to:

$$E(i_1) := \left\{ \boldsymbol{y}' \in \mathcal{Y} \ : \ \arg\max_{\boldsymbol{z} \in \mathcal{Z}} s(\boldsymbol{z}, \boldsymbol{y}') \in M_{i_1} \right\} \tag{13}$$

We use this simplified case to introduce our null hypotheses and test statistics attached to this selection event, and consider two options:

- A first option consists in representing the mesh $M_{i_1}$ by its center $\boldsymbol{c}_1$. Then the corresponding null hypothesis is the following:

$$\mathbb{H}'_{0,1} \ : \ \text{``} s(\boldsymbol{c}_1, \boldsymbol{\mu}) = 0\text{''}, \tag{14}$$

  It can be tested using statistic $V'_1 = s(\boldsymbol{c}_1, \boldsymbol{y})$.

- A second possibility is to represent $M_{i_1}$ by the motif with the highest association score within. In this case, the null hypothesis becomes:

$$\mathbb{H}''_{0,1} \ : \ \text{``} \forall \boldsymbol{z} \in M_{i_1} \ , \ s(\boldsymbol{z}, \boldsymbol{\mu}) = 0\text{''}, \tag{15}$$

  We test it using statistic $V''_1 = \max_{\boldsymbol{z} \in M_{i_1}} s(\boldsymbol{z}, \boldsymbol{y})$.

---

**Algorithm 2** Hypersphere Directions hit-and-run sampler

---

```
/* Description:  The Hypersphere Directions hit-and-run sampler creates a discrete-time
   Markov chain on an open and bounded region and is used to approximate a uniform
   distribution on the selection event E.                                            */
```

**Inputs:** *Response* $\boldsymbol{y} \in E \subseteq \mathbb{R}^n$, *B and R the numbers of burn-in iterations and replicates.*

**Result:** $\boldsymbol{y}'^{(B+1)}, ..., \boldsymbol{y}'^{(B+R)} \in E \subseteq \mathbb{R}^n$ the replicates sampled under the conditional null distribution

10   $\tilde{\boldsymbol{y}}'^{(0)} \leftarrow \mathbb{L}(\boldsymbol{y})$; /* $\mathbb{L}$ is the cumulative distribution function of $\mathcal{N}(\boldsymbol{\mu}, \sigma^2 \boldsymbol{C}_n)$   */

11   **for** $t = 1, \ldots, B + R$ **do**

12     Sample uniformly $\boldsymbol{\theta}^{(t)}$ from $\{\boldsymbol{\theta} \in \mathbb{R}^n, \ \|\boldsymbol{\theta}\| = 1\}$;

13     $a^{(t)} \leftarrow \max \left\{ \max\limits_{\boldsymbol{\theta}_t^{(i)} > 0} -\dfrac{\tilde{\boldsymbol{y}}'^{(t-1)}}{\boldsymbol{\theta_t}}; \ \max\limits_{\boldsymbol{\theta}_t^{(i)} < 0} \dfrac{1 - \tilde{\boldsymbol{y}}'^{(t-1)}}{\boldsymbol{\theta_t}} \right\}$;

14     $b^{(t)} \leftarrow \max \left\{ \min\limits_{\boldsymbol{\theta}_t^{(i)} < 0} -\dfrac{\tilde{\boldsymbol{y}}'^{(t-1)}}{\boldsymbol{\theta_t}}; \ \min\limits_{\boldsymbol{\theta}_t^{(i)} > 0} \dfrac{1 - \tilde{\boldsymbol{y}}'^{(t-1)}}{\boldsymbol{\theta_t}} \right\}$;   /* Sampling $\lambda^{(t)}$ from $]a^{(t)}, b^{(t)}[$ ensures that

      $\tilde{\boldsymbol{y}}'^{(t-1)} + \lambda^{(t)} \boldsymbol{\theta^{(t)}} \in ]0, 1[^n$                                        */

15     **while** $\boldsymbol{y}'^{(t)} \notin E$ **do**

```
            /* This loop is parallelized on several cores until one of them discovers a
               replicates in the selection event.                                       */
```

16       Sample uniformly $\lambda^{(t)}$ from $]a^{(t)}, b^{(t)}[$;

17       $\tilde{\boldsymbol{y}}'^{(t)} \leftarrow \tilde{\boldsymbol{y}}'^{(t-1)} + \lambda^{(t)} \boldsymbol{\theta^{(t)}}$;

18       $\boldsymbol{y}'^{(t)} \leftarrow \mathbb{L}^{-1}(\tilde{\boldsymbol{y}}'^{(t)})$;

19     **end**

20   **end**

---

In both cases, we reject the null hypothesis if the test statistics are greater than a threshold, determined by their cumulative distributions under the nulls (14), (15) conditionally to $E(i_1)$ : $\mathbb{F}'_{1,(i_1)}$ and $\mathbb{F}''_{1,(i_1)}$. In practice, there is no closed form for these conditional cumulative distributions, and we rely on an empirical version that we build using a hit-and-run sampler algorithm, as described in Section 4.4.

Hypotheses (14) and (15) lead to very similar results when the meshes are small enough, which is easily the case in practice. (14) gives us insights on one specific motif of the mesh — the center, but (15) tells us about whether there exists a motif within $M_{i_1}$ associated with the phenotype. To illustrate the difference, let us consider a meshing with only one bin per coordinate, that is the meshing with only one mesh, containing all the motifs:

- Testing the center-based null hypothesis (14) boils down to testing the association of $\boldsymbol{\mu}$ with the motif $\boldsymbol{c}_1$ with the same probabilities for each letter of $\mathcal{A}$ at every position, and produces a $p$-value of 1, regardless of the data, since for any $k$-mer $u$, $\|\boldsymbol{c} - \boldsymbol{u}\|^2 = k \times (0.75^2 + 3 \times 0.25^2)$, which leads to $\forall \boldsymbol{X} \in \mathcal{X}, \ \boldsymbol{\varphi}^{\boldsymbol{c}, \boldsymbol{X}} = \boldsymbol{0}$ according to the centering step, and to a zero score for any $\boldsymbol{y}' \in \mathcal{E}$ .

- By contrast, one can obtain a strictly less than 1 $p$-value for (15), because different $\boldsymbol{y}' \in \mathcal{E}$ can lead to different scores, which means that there may exist a motif in $\mathcal{Z}$ associated with $\boldsymbol{y}$ — but does not inform us on which motif it is.

### 4.4   Sampling from the conditional null distribution with the Hit-and-Run algorithm

Even after reducing our selection to a finite set (Section 4.2), a rejection sampling strategy that would draw $\boldsymbol{y}'$ from either (9, 16) or (9, 17) and only retain those leading to the selection of the same mesh as $\boldsymbol{y}$ is not tractable as the rejection rate is empirically too low. Following Slim et al. (2019), we resort to a Hypersphere Direction strategy (Algorithm 2).

The hit-and-run algorithm produces uniform samples from an open and bounded acceptance region—corresponding, in our case, to the selection event. It starts from any point in the acceptance region, draws a random direction from this point and samples along this direction until it finds one elements that also falls in the acceptance region. It then follows the same procedure from this new starting point. The hit-and-run sampler therefore also relies on rejection but it does so along a single dimension rather than from $\mathbb{R}^n$. It explores the selection event step by step, starting from a point that belongs to this event, which guarantees a higher acceptance rate. To speed up the procedure, we parallelize the rejection step across several cores. Because each point sampled by the hit-and-run procedure depends on the previous one, it is impossible to parallelize the whole sampling process. By contrast, the rejection step used for computing a single replicate, once a sampling direction has been fixed, can be parallelized. We draw several distances to the initial point independently, optimizing new independent points, until one of them belongs to the selection event. This parallelization provides a significant time saving, as discussed in Section 5.3. Algorithm 2 produces uniform samples from an open and bounded acceptance region. The boundedness assumption does not hold in our case as the $\arg\max$ over $\mathcal{Z}$ of the score only depends on the direction of $\boldsymbol{y}$ and not on its norm. The openness requirement is ensured by the definition of the meshes. Following Slim et al. (2019) again, we use the reparameterization $\tilde{\boldsymbol{y}} = \mathbb{L}(\boldsymbol{y})$, where $\mathbb{L} : \mathbb{R}^n \to ]0, 1[^n$ is defined as $\mathbb{L}(\boldsymbol{y})_i = \mathbb{L}_{\boldsymbol{\mu},\sigma^2}(\boldsymbol{y}_i)$ for $i = 1, \dots, n$ and $\mathbb{L}_{\boldsymbol{\mu},\sigma^2}$ denotes the cumulative distribution function of $\mathcal{N}(\boldsymbol{\mu}, \sigma^2 \boldsymbol{C}_n)$. Sampling uniform $\tilde{\boldsymbol{y}}$ from the open bounded space $]0, 1[^n$ indirectly provides normal samples from $\mathcal{N}(\boldsymbol{\mu}, \sigma^2 \boldsymbol{C}_n)$.

Combining this sampling strategy with the quantization of the selection event introduced in Section 4.2 and the selective null hypotheses attached to this event introduced in Section 4.3 provides a selective inference procedure for one selected motif $\boldsymbol{z}_1$ ($q = 1$) and a null defined by a given pair $(\boldsymbol{\mu}, \sigma)$ of parameters. Our next two steps are to handle the selection of multiple motifs, and the general case where several $\boldsymbol{\mu}$ describe the same null hypothesis and $\sigma$ is not specified.

### 4.5 Dealing with the selection of several motifs ($q > 1$)

We now consider that we selected $q > 1$ motifs with the SEISM procedure, leading to the general (12) selection event $E(i_1, \dots, i_q)$ . Generalizing our single-motif strategy of Section 4.3, we propose two options for defining null hypotheses (and test statistics) related to this selection event:

- The first one relies on the centers of the selected meshes:

$$\mathbb{H}_{0,j} \; : \; \text{``}s(\boldsymbol{c}_j, \boldsymbol{\Pi}'_j \boldsymbol{\mu}) = 0\text{''}, \tag{16}$$

  where $\boldsymbol{\Pi}'_j$ is the orthogonal projector onto $\mathrm{Span}_{\ell \neq j} \left\{ \boldsymbol{\varphi}^{\boldsymbol{c}_\ell, \boldsymbol{X}} \right\}^{\perp}$. In other words, it expresses that the center of the mesh $M_{i_j}$ is associated with $\boldsymbol{\mu}$ after removing its component carried by the span of the centers of the meshes corresponding the the $q - 1$ other motifs.

- And the second one takes advantages of the best motifs in each mesh:

$$\mathbb{H}_{0,j} \; : \; \text{``}\forall (\boldsymbol{z}^*_{i_\ell})_{\ell \neq j} \in (M_{i_\ell})_{\ell \neq j}, \quad \forall \boldsymbol{z} \in M_{i_j}, \; s(\boldsymbol{z}, \boldsymbol{\Pi}'' \left( (\boldsymbol{z}^*_{i_\ell})_{\ell \neq j} \right) \boldsymbol{\mu}) = 0\text{''}, \tag{17}$$

  with $\boldsymbol{\Pi}'' \left( (\boldsymbol{z}^*_{i_\ell})_{\ell \neq j} \right)$ being the projection onto $\mathrm{Span}_{\ell \neq j} \left\{ \boldsymbol{\varphi}^{\boldsymbol{z}^*_{i_\ell}, \boldsymbol{X}} \right\}^{\perp}$.

Generalizing what we introduced for $q = 1$ (Section 4.3), we test those hypotheses using $V'_j = s(\boldsymbol{c}_j, \boldsymbol{\Pi}'_j \boldsymbol{y})$ and $V''_j = \max_{\boldsymbol{z} \in M_{i_j}} s(\boldsymbol{z}, \boldsymbol{\Pi}''_j \boldsymbol{y})$. To that end, we rely on their cumulative distributions under the nulls (16), (17) conditionally to $E(i_1, \dots, i_q)$ : respectively $\mathbb{F}'_{1,\dots,q(i_1,\dots,i_q)}$ and $\mathbb{F}''_{1,\dots,q,(i_1,\dots,i_q)}$, empirically approximated with Algorithm 2.

Following the work of Loftus & Taylor (2015) in the finite case, both versions of our null hypothesis are joint across the $q$ motifs: each of them considers the association between the $j$-th selected motif and $\boldsymbol{\mu}$ after projecting onto the span of all others, not just the ones that were selected before — using $\boldsymbol{\Pi}'$ and $\boldsymbol{\Pi}''$. This is to be contrasted to our sequential selection process, which adjusts at each steps for the previously selected motifs using $\boldsymbol{P}$.

In order to give more insights on these null hypotheses, we derive the following proposition:

**Proposition 4.1** (Description of the selective nulls). *Let $\boldsymbol{Z} = \{\boldsymbol{z}_1, \ldots, \boldsymbol{z}_q\}$ be $q$ sequence motifs. Let $s(\cdot, \cdot)$ be a score such that "nullity implies orthogonality" (for instance $s^{HSIC}$ or $s^{ridge}$):*

$(\mathbf{A_1})$ **Nullity implies orthogonality:** *If $\{s(\boldsymbol{z}, \boldsymbol{y}) = 0\}$ then $\{\langle \boldsymbol{\varphi}^{\boldsymbol{z}, \boldsymbol{X}}, \boldsymbol{y} \rangle = 0\}$, for every $(\boldsymbol{y}, \boldsymbol{z}) \in \mathcal{E} \times \mathcal{Z}$, and for some function $\boldsymbol{z} \to \boldsymbol{\varphi}^{\boldsymbol{z}, \boldsymbol{X}} \in \mathcal{E}$.*

*Let $\boldsymbol{\mu} \in \mathcal{E}$ and decompose $\boldsymbol{\mu}$ as*

$$\boldsymbol{\mu} = \sum_{j=1}^{q} \alpha_j \boldsymbol{\varphi}^{\boldsymbol{z}_j, \boldsymbol{X}} + \underline{\boldsymbol{\mu}} \tag{18}$$

*with $\underline{\boldsymbol{\mu}} \in \mathcal{E}$ orthogonal to $\mathrm{Span}(\boldsymbol{\varphi}^{\boldsymbol{Z}, \boldsymbol{X}})$.*

*It holds that "$s(\boldsymbol{z}_j, \boldsymbol{\Pi_j} \boldsymbol{\mu}) = 0$" is equivalent to "$\alpha_j = 0$" for some decomposition (18).*

If $\mathrm{Rank}(\boldsymbol{\varphi}^{\boldsymbol{Z}, \boldsymbol{X}}) = q$ then the decomposition (18) is unique, and the greedy selection procedure described in Section 3 enforces this situation. We interpret this as follows: we look at a motif $\boldsymbol{z}_\ell$ and would like to test its significance; in view of property $(\mathbf{A_1})$, we can eliminate the effects that are captured by the other motifs by using the orthogonal projection onto the orthogonal of $\mathrm{Span}(\boldsymbol{\varphi}^{\boldsymbol{z}_j, \boldsymbol{X}})$, given by $\boldsymbol{\Pi}_j$ (using $\boldsymbol{\Pi}_j = \boldsymbol{\Pi}'_j$ or $\boldsymbol{\Pi}_j = \boldsymbol{\Pi}'' \left( (\boldsymbol{z}^*_{\boldsymbol{i_\ell}})_{\ell \neq j} \right)$), and consider $\boldsymbol{\Pi_j y}$ to test the association "$s(\boldsymbol{z}_j, \boldsymbol{\Pi_j \mu}) = 0$"; equivalent to testing "$\alpha_j = 0$" by the above proposition.

### 4.6 Sampling under selective multiple hypotheses with known $\sigma$

The sampling strategy described in Section 4.4 builds a conditional null distribution—therefore offering a selective inference procedure—for a given $\boldsymbol{\mu}$ and $\sigma$. In practice, $\sigma$ is not known, and several values of $\boldsymbol{\mu}$ can describe the selective null hypotheses (16) or (17) for a given motif selection. Of note, this issue is not specific to our selective inference procedure. It will arise in any sampling-based post-selection inference strategy including data-split: even if the latter samples from a non-selective null hypothesis, it still needs specific values for $\boldsymbol{\mu}$ and $\sigma$. For $\boldsymbol{\mu}$, the issue is that any fixed value can not represent the whole set of possible values, which would modify the null hypothesis actually tested. For the variance parameter, it may be fixed by the user, but this may lead to a non-valid procedure if the chosen value is different from the real one.

We leave aside the choice of $\sigma$ for now, and describe how we can sample from any null distribution (16) or (17) using $\boldsymbol{\mu} = 0$ for a given $\sigma$. Our results holds for scores verifying the following assumption—this includes both $s^{\mathrm{HSIC}}$ and $s^{\mathrm{ridge}}$:

$(\mathbf{A_2})$ **Nullity implies translation-invariant:** *If $s(\boldsymbol{z}, \boldsymbol{y}) = 0$ then $\forall \boldsymbol{y}' \in \mathcal{E}$, $s(\boldsymbol{z}, \boldsymbol{y}') = s(\boldsymbol{z}, \boldsymbol{y} + \boldsymbol{y}')$, for every $(\boldsymbol{y}, \boldsymbol{z}) \in \mathcal{E} \times \mathcal{Z}$;*

Under this assumption, the following proposition ensures that using the quantile of the empirical distribution of scores sampled under $\boldsymbol{\mu} = 0$ leads to a calibrated test procedure:

**Proposition 4.2.** *Let $s$ be an association score such that $(\mathbf{A_2})$ holds. Let $V'_j = s(\boldsymbol{c}_j, \boldsymbol{\Pi}'_j \boldsymbol{y})$ and $V''_j = \max_{\boldsymbol{z} \in M_{i_j}} s(\boldsymbol{z}, \boldsymbol{\Pi}''_j \boldsymbol{y})$, formed from $\boldsymbol{y}$ sampled from (9) with any mean $\boldsymbol{\mu}$ such that $s(\boldsymbol{z}', \boldsymbol{\mu}) = 0$, any known variance $\sigma > 0$, and such that $\boldsymbol{z}' = \arg\max_{\boldsymbol{z} \in \mathcal{Z}} s(\boldsymbol{z}, \boldsymbol{y})$. The conditional null distributions $\mathbb{F}'_{j, (i_1, \ldots, i_q)}$ and $\mathbb{F}''_{j, (i_1, \ldots, i_q)}$, with mean $\boldsymbol{0}$ and variance $\sigma$ verify:*

$$\mathbb{F}'_{j, (i_1, \ldots, i_q)}(V'_j) \sim Unif(0, 1) \text{ and } \mathbb{F}''_{j, (i_1, \ldots, i_q)}(V''_j) \sim Unif(0, 1)$$

*Proof.* Assumption $(\mathbf{A_2})$ under the Gaussian model (9) implies the following property:

$$\begin{aligned} \forall (\boldsymbol{z}, \boldsymbol{A}, \boldsymbol{y}) \in \mathcal{Z} \times \boldsymbol{A} \times \mathcal{E} \text{ such that } \boldsymbol{y} = \boldsymbol{\mu} + \sigma \boldsymbol{\epsilon}, \\ \text{"}s(\boldsymbol{z}, \boldsymbol{A}\boldsymbol{\mu}) = 0\text{"} \implies \text{"}s(\boldsymbol{z}, \boldsymbol{A}\boldsymbol{y}) = s(\boldsymbol{z}, \sigma \boldsymbol{A} \varepsilon)\text{"}, \end{aligned} \tag{19}$$

which implies that, for a composite null hypothesis of the form $\mathbb{H}_0$ : "$s(\boldsymbol{z}, \boldsymbol{A}\boldsymbol{\mu}) = 0$", the distribution of $s(\boldsymbol{z}, \boldsymbol{A}\boldsymbol{y})$ does not depend on the mean $\boldsymbol{\mu}$ that satisfies $\mathbb{H}_0$. Hence, even if the hypothesis $\mathbb{H}_0$ corresponds to a set of probability distributions of $\boldsymbol{y}$ that may depend on $\boldsymbol{\mu}$, the distribution of the statistic $s(\boldsymbol{z}, \boldsymbol{A}\boldsymbol{y})$ does not depend on $\boldsymbol{\mu}$ under this hypothesis. We can then conclude that if $\sigma$ is known, as it is assumed to be the case in this section, then a test statistic of the form $V = s(\boldsymbol{z}, \boldsymbol{\Pi}\boldsymbol{y})$ has the same distribution as $s(\boldsymbol{z}, \sigma\boldsymbol{\Pi}\boldsymbol{\epsilon})$. □

### 4.7 Sampling under selective multiple hypotheses with unknown $\sigma$

In practice, $\sigma$ is often unknown. To address this issue, we rely on the normalized versions of the test statistics $V'$ and $V''$ introduced in Section 4.3, defined by

$$T'_j := \frac{s(\boldsymbol{c}_j, \boldsymbol{\Pi}'_j \boldsymbol{y})}{\|\boldsymbol{y}\|^2} \quad \text{and} \quad T''_j := \max_{\boldsymbol{z} \in M_{i_j}} \frac{s\Big(\boldsymbol{z}, \boldsymbol{\Pi}''\big((\boldsymbol{z}_\ell)_{\ell \neq j}\big)\boldsymbol{y}\Big)}{\|\boldsymbol{y}\|^2} \tag{20}$$

where $\boldsymbol{z}_\ell = \arg\max_{\boldsymbol{z} \in \mathcal{Z}} s(\boldsymbol{z}, \boldsymbol{P}_\ell \boldsymbol{y})$. We will denote $\mathbb{G}'_{j,(i_1,\dots,i_q)}$ and $\mathbb{G}''_{j,(i_1,\dots,i_q)}$ their cumulative distribution functions under the null, conditionally to $E(i_1, \dots, i_q)$.

We will also make use of a third assumption, here again fulfilled by $s^{\text{HSIC}}$ and $s^{\text{ridge}}$:

($\mathbf{A_3}$) **Two-homogeneity:** It holds that $s(\boldsymbol{z}, t\boldsymbol{y}) = t^2 s(\boldsymbol{z}, \boldsymbol{y})$ for all $(\boldsymbol{y}, \boldsymbol{z}) \in \mathcal{E} \times \mathcal{Z}$ and all $t > 0$.

Of note, normalizing the association score with respect to the labels does not affect the selection:

$$\forall \boldsymbol{y} \in \mathcal{Y}, \quad \arg\max_{\boldsymbol{z} \in \mathcal{Z}} s(\boldsymbol{z}, \boldsymbol{y}) = \arg\max_{\boldsymbol{z} \in \mathcal{Z}} \frac{s(\boldsymbol{z}, \boldsymbol{y})}{\|\boldsymbol{y}\|^2} \tag{21}$$

If $\boldsymbol{\mu} = \boldsymbol{0}$, the distribution of the normalized statistics does not depend on $\sigma$, and the empirical cumulative distribution functions of normalized scores obtained by sampling under $\boldsymbol{\mu} = \boldsymbol{0}$ and any $\sigma$ still provide a valid inference procedure :

**Proposition 4.3.** *Let $s$ be an association score such that $(\mathbf{A_2})$ and $(\mathbf{A_3})$ hold. Let $T'_j = s(\boldsymbol{c}_j, \boldsymbol{\Pi}'_j \boldsymbol{y})/\|\boldsymbol{y}\|^2$ and $T''_j = \max_{\boldsymbol{z} \in M_{i_j}} s(\boldsymbol{z}, \boldsymbol{\Pi}''_j \boldsymbol{y})/\|\boldsymbol{y}\|^2$, formed from $\boldsymbol{y}$ sampled from (9) with mean $\boldsymbol{\mu} = \boldsymbol{0}$, and any variance $\sigma > 0$. Then for all $\sigma' > 0$, their conditional null distributions $\mathbb{G}'_{j,(i_1,\dots,i_q)}$ and $\mathbb{G}''_{j,(i_1,\dots,i_q)}$ with mean $\boldsymbol{0}$ and variance $\sigma'$ verify:*

$$\mathbb{G}'_{j,(i_1,\dots,i_q)}(T'_j) \sim Unif(0,1) \ and \ \mathbb{G}''_{j,(i_1,\dots,i_q)}(T''_j) \sim Unif(0,1)$$

*Proof.* Let us consider two different normal models as defined in (9) under the global null hypothesis "$\boldsymbol{\mu} = \boldsymbol{0}$" and given by

$$y^{(1)} = \sigma^{(1)}\boldsymbol{\varepsilon}^{(1)} \quad \text{and} \quad y^{(2)} = \sigma^{(2)}\boldsymbol{\varepsilon}^{(2)}$$

Then

$$\frac{s(\boldsymbol{c}_j, \boldsymbol{\Pi}'_j \boldsymbol{y}^{(1)})}{\|\boldsymbol{y}^{(1)}\|^2} \sim \frac{s(\boldsymbol{c}_j, \boldsymbol{\Pi}'_j \boldsymbol{y}^{(2)})}{\|\boldsymbol{y}^{(2)}\|^2} \quad \text{and} \quad \frac{s\big(\boldsymbol{z}, \boldsymbol{\Pi}''\left((\boldsymbol{z}_\ell)_{\ell \neq j}\right) \boldsymbol{y}^{(1)}\big)}{\|\boldsymbol{y}^{(1)}\|^2} \sim \frac{s\big(\boldsymbol{z}, \boldsymbol{\Pi}''\left((\boldsymbol{z}_\ell)_{\ell \neq j}\right) \boldsymbol{y}^{(2)}\big)}{\|\boldsymbol{y}^{(2)}\|^2}.$$

The proof directly follows assumption $(\mathbf{A_3})$ applied with $t = \|\boldsymbol{y}^{(\cdot)}\|^2$. Proposition 4.3 is complementary to Proposition 4.2 and provides a selective inference procedure when $\sigma$ is unknown, under the special null hypothesis $\boldsymbol{\mu} = 0$. □

Our final result investigates the testing procedures for the general null hypotheses (16) and (17)—not restricted to $\boldsymbol{\mu} = \boldsymbol{0}$—with an unknown $\sigma$. Recall that the decision rule is to reject the null hypothesis if the observed value of the statistic is greater than a given threshold $t$. We show that choosing $t$ to be a quantile for the global null hypothesis ($\boldsymbol{\mu} = 0$) leads to a calibrated (for the type I error) non-selective procedure, see (22).

**Proposition 4.4** (Global null achieves lowest observed significance). *Let $\boldsymbol{Z} = \{\boldsymbol{z}_1, \ldots, \boldsymbol{z}_q\}$ be q sequence motifs. Let $s(\cdot, \cdot)$ be a score such that $(\mathbf{A_1})$ and $(\mathbf{A_2})$ hold. Let $\boldsymbol{\mu} \in \mathcal{E}$ be such that*

$$\mathbb{H}_0 : \text{``}s(\boldsymbol{Z}, \boldsymbol{\mu}) = 0\text{''}$$

*Then*

$$\forall t > 0, \quad \sup_{\boldsymbol{\mu} \in \mathbb{H}_0} \mathbb{P}\left[\frac{s(\boldsymbol{Z}, \boldsymbol{\mu} + \sigma \boldsymbol{\epsilon})}{\|\boldsymbol{\mu} + \sigma \boldsymbol{\epsilon}\|^2} \geq t\right] = \mathbb{P}\left[\frac{s(\boldsymbol{Z}, \boldsymbol{\epsilon})}{\|\boldsymbol{\epsilon}\|^2} \geq t\right] \tag{22}$$

We provide a proof in Appendix C. This proof makes an ad-hoc use of Anderson's theorem on a symmetric convex cone (whereas it is usually devoted to symmetric convex bodies).

Proposition 4.4 shows that data-split produces a calibrated procedure for testing the general null hypotheses (16) and (17) when sampling under the global null ($\boldsymbol{\mu} = 0$) the test statistics (20). We could not prove an equivalent statement for conditional null hypotheses, and Proposition 4.4 therefore does not guarantee the validity of a selective inference procedure sampling under the global null ($\boldsymbol{\mu} = 0$). Yet, we used it as a heuristic justification of SEISM and we observed that it leads to empirically calibrated procedures, see Section 5.2.

In view of Proposition 4.4 and its proof, one can see that the alternatives $\boldsymbol{\mu}$ such that $\|\boldsymbol{P}_q \boldsymbol{\mu}\|/\|\boldsymbol{\mu}\|$ is large have small power. As the selection procedure described in Section 3 achieves good results (Section 5.1), the chosen motifs $\boldsymbol{Z}$ should capture the principal components of $\boldsymbol{\mu}$, and therefore are such that $\|\boldsymbol{P}_q \boldsymbol{\mu}\|/\|\boldsymbol{\mu}\|$ should be small.

## 5 Results

### 5.1 SEISM performs as well as state-of-the-art *de novo* motif discovery methods

In order to compare the accuracy of our selection step with existing motif discovery algorithms, we use the 40 ENCODE Transcription Factors ChIP-seq datasets from K562 cells (ENCODE Project Consortium, 2004), each of which contains a known TF motif, denoted $\boldsymbol{m}^*$, derived using completely independent assays (Jolma et al., 2013). STREME (Bailey, 2021) and MEME (Bailey et al., 2006) are state-of-art bioinformatics methods for *de-novo* motifs discovery tasks. STREME identifies motifs that maximize a Fisher score of association between the presence of the motif and the binary class of sequences. By looking for maximum likelihood estimates of the parameters of a mixture model - made up of a background distribution and a model for generating $k$-mers at some positions - that may have produced a particular dataset using an expectation maximisation technique, MEME finds enriched motifs in this dataset. Finally CKN-seq (Chen et al., 2017) is a one-layer CNN tailored to small scale datasets. We set up STREME, MEME and SEISM to select 5 sequence motifs. SEISM is run with a regularization parameter $\lambda = 0.01$. CKN-seq jointly optimizes its filters, which notoriously leads to poor performances when few filters are used. We train it consequently over 128 filters. We measure these accuracy of all methods by comparing the motifs they discover with the known motif corresponding to the transcription factor $\boldsymbol{m}^*$. We rely on the Tomtom method (Gupta et al., 2007), which quantifies the probability that the euclidean distance between a random motif and $\boldsymbol{m}^*$ is lower than the distance between the discovered motif and $\boldsymbol{m}^*$. More precisely for each method we use the lowest Tomtom $p$-value between the known TF motif $\boldsymbol{m}^*$ and any of those discovered by the method. The Tomtom score is then defined as $-\log_{10}$ of the Tomtom $p$-value. We define the accuracy of the method as the proportion of experiments where the Tomtom score between its best match and the true TF motif was higher than some threshold.

Figure 4 (left panel) demonstrates that SEISM is just as good as, if not superior to, state-of-the-art bioinformatics algorithms at detecting sequence motifs when thresholding Tomtom p-values at 0.01. The one-layer CNN with jointly optimized filters performs poorly in this experiments, emphasising the importance of greedy optimization for selecting the right motif.

Figure 4 (right panel) shows that SEISM performs slightly worse than STREME and MEME for high thresholds on the Tomtom scores. This suggests that the matrix $z$ that SEISM identifies is close enough to the PWM matrix of the true motif, but farther away than the matrices identified by STREME or MEME.

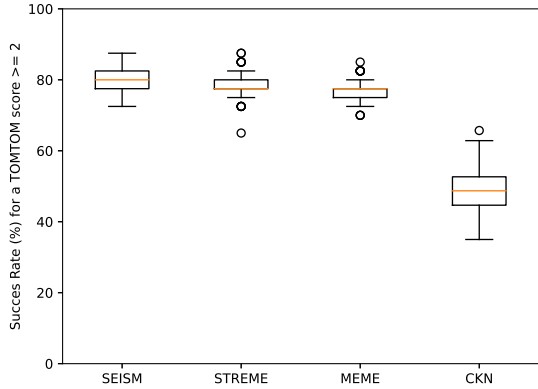
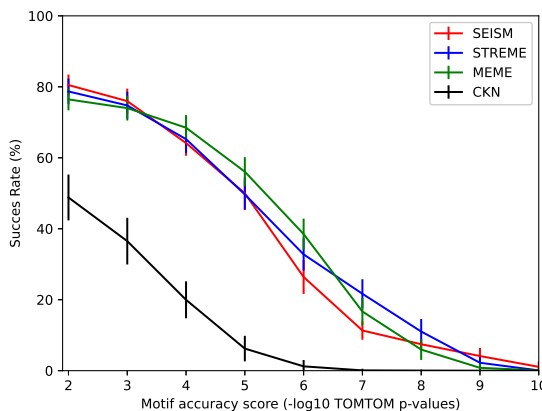

Figure 4: **Left**: Proportion of datasets where the true motif was detected by the designated algorithm. A true motif is said to be detected if its highest Tomtom score with the discovered motifs is greater than 2. **Right**: Accuracy of motif discovery algorithms on ENCODE TF ChIP-seq datasets. The curves display the proportion of ChIP-seq datasets were the best motif identified by the designated algorithm has a Tomtom score greater than $x$.

This discrepancy reflects a different usage of $z$ to parameterize a distribution of $k$-mers. In practice, we observe that on a given dataset, the $p$-values of the best motifs discovered by SEISM and STREME are not separated by more than 2 orders of magnitude, which leads to minor differences in the motifs, as illustrated in Table 1.

| Reference motif
*(ATF4_DBD)* | SEISM motif
*p-value*$= 10^{-6}$ | STREME motif
*p-value*$= 3 \times 10^{-8}$ |
| :---: | :---: | :---: |
| 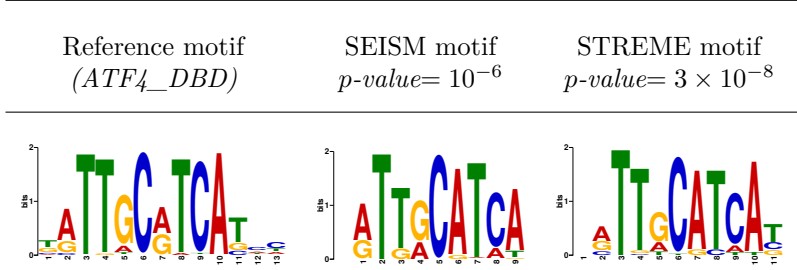 | | |

Table 1: Comparison between two discovered sequence motifs by SEISM or STREME, and the true motif *(ATF4_DBD)*

Both SEISM and MEME/STREME exploit a distribution of $k$-mers at the transcription factor binding site. MEME and STREME maximize the likelihood of a *categorical model*, whereby the matrix $\boldsymbol{z}$ directly defines the probability to observe each letter at each of the $k$ sites:

$$\forall (\boldsymbol{u}, \ \boldsymbol{z} \in \mathcal{Z}), \quad \mathcal{L}_{\text{cat}}(\boldsymbol{u}; \boldsymbol{z}) = \prod_{i=1}^{k} \boldsymbol{u}_i^T \boldsymbol{z}_i \tag{23}$$

SEISM on the other hand is based on a *Gaussian model*. Through representation (3), $\boldsymbol{z}$ is meant to maximize the Gaussian likelihood of a set of $k$-mers, *i.e.*

$$\forall (\boldsymbol{u}, \ \boldsymbol{z} \in \mathcal{Z}), \quad \mathcal{L}_{\text{gaus}}(\boldsymbol{u}; \boldsymbol{z}) = C \prod_{i=1}^{k} e^{-\frac{\|\boldsymbol{u}_i - \boldsymbol{z}_i\|^2}{2\omega^2}} \tag{24}$$

where $C$ is a constant such that the sum of probabilities over $\mathbb{R}^{4 \times k}$ equals 1. If we consider a binary $\boldsymbol{y}$ to match the setting of MEME/STREME, this set is made of one $k$-mer for each positive sequence. Importantly,

the true TF motifs from (Jolma et al., 2013) that we use to assess selection accuracies are also defined through the maximum likelihood in a categorical model, which can explain why the $z$ obtained with MEME/STREME are closer to the true PWM than the one obtained with SEISM.

We now illustrate on a simple example how the same distribution of $k$-mers is parameterized by different matrices under the two models. To build an easy example, we focus on $k$-mers of length 1, with

$$P(A) = 0.3, \; P(C) = 0.4, \; P(G) = 0.1, \; P(T) = 0.2 \tag{25}$$

The matrix $\boldsymbol{z}_1 = (0.3, 0.4, 0.1, 0.2)^T$ used with the categorical model trivially constructs such a distribution. But using the same matrix in a Gaussian model with a parameter $\omega$ fixed as described in Appendix A leads to a slightly different distribution:

$$P(A) = 0.28, \; P(C) = 0.43, \; P(G) = 0.11, \; P(T) = 0.18 \tag{26}$$

A distribution closer to Equation (25) can be constructed with a Gaussian model parameterized by $\boldsymbol{z}_2 = (0.315, 0.38, 0.08, 0.225)^T$.

To clarify the relationships between those two motifs, we will rewrite (23) considering $\boldsymbol{u}$ is one hot encoded. That is, for each position $i$, it has only one 1 for letter $j(i)$ and 0's elsewhere:

$$\forall \left( \boldsymbol{u}, \; \boldsymbol{z} \in \mathcal{Z} \right), \quad \mathcal{L}_{\text{cat}}(\boldsymbol{u}; \boldsymbol{z}) = \prod_{i=1}^{k} \boldsymbol{z}_{i, j(i)} \tag{27}$$

Assuming that the columns of $\boldsymbol{z}$ are normalized and $\omega = 1$, we can modify (24):

$$\forall \left( \boldsymbol{u}, \; \boldsymbol{z} \in \mathcal{Z} \right), \quad \mathcal{L}_{\text{gaus}}(\boldsymbol{u}; \boldsymbol{z}) = C \prod_{i=1}^{k} e^{-\frac{\|\boldsymbol{u}_i - \boldsymbol{z}_i\|^2}{2\omega^2}} = C_2 \prod_{i=1}^{k} e^{\boldsymbol{u}_i^T \boldsymbol{z}_i} = C_2 \prod_{i=1}^{k} e^{\boldsymbol{z}_{i, j(i)}} \tag{28}$$

With the Gaussian model and a few assumptions, the motifs can the be seen as defining the log probability to observe each letter at each of the $k$ sites. This gives us a new interpretation for the filters learned by CNNs and suggests that in this framework it might be interesting to constrain $e^{\boldsymbol{z}}$ rather than $\boldsymbol{z}$ to be in $\mathcal{Z}$.

We used a Gaussian activation function since it is closer to typical CNNs approaches. Our framework is generic enough to allow other activation functions based on the categorical model, or more realistic variants (Ruan & Stormo, 2017).

## 5.2 Statistical validity and performances

In order to assess the statistical validity and of the SEISM procedure with the different strategies, we simulate datasets under the null hypothesis. To that end, we draw one sequence motif $\tilde{\boldsymbol{z}}$ with length $k = 8$ for each simulated dataset using a uniform distribution on $\mathcal{Z}$ restricted to motifs with an information level fixed at 10 bits. Then, we draw a set of $n = 30$ biological sequences $X$ as follows: all sites are generated according to a uniform distribution over A, C, T, G for all sequences, and for half of the sequences one $k$-mer is drawn according to the categorical model parameterized by $\tilde{\boldsymbol{z}}$. The phenotypes $\boldsymbol{y}$ are drawn from $\mathcal{N}(\boldsymbol{0}, \sigma^2 \boldsymbol{C}_n)$ to generate data under the null hypothesis for calibration experiments, and from $\mathcal{N}(\boldsymbol{\varphi}^{\tilde{\boldsymbol{z}}, \boldsymbol{X}}, \sigma^2 \boldsymbol{C}_n)$ to generate data under the alternative for experiments on statistical power, with $\sigma = 0.1$ in both cases. We then run the SEISM procedure to select and test two sequence motifs. For both the data-split strategy and the hypersphere direction sampling one, the distribution from which the replicates are drawn uses the empirical variance from $\boldsymbol{y}$ as variance parameter. Although any choice for this parameter leads to a valid procedure, as described in Section 4.6, we make this choice for numerical stability considerations. For the data-split strategy, we sample 1000 replicates under the null hypothesis to compute the $p$-value. For SEISM, we sample $50,000$ replicates under the conditional null hypothesis using the hypersphere direction sampler, after $10,000$ burn-in iterations.

Figure 5 (top) shows the Q-Q plot of the distribution of quantiles of the uniform distribution against the $p$-values obtained across 1000 datasets under the null hypothesis for the data-split strategy and 100 datasets

for the hypersphere direction sampling one. All the data points are well-aligned with the diagonal, which confirms the correct calibration of both the data-split and hypersphere direction sampling strategies, either considering the best motif or the center of the mesh and regardless of the size parameter.

Figure 5 (bottom) shows the same Q-Q plot on data generated under the alternative hypothesis. From this figure, we observe that on small datasets, the post-selection strategy is more powerful than the data-split one, regardless of the size of the mesh considered or the choice concerning the definition of the null hypothesis. The variance observed on the curves associated with the selective inference procedure is due to the presence of a weak residual signal after the first motif as a result of an imperfect selection step. Testing it with the best motif in the mesh captures this signal, resulting in curves under the diagonal. By contrast, focusing on the center of the meshes leads to testing motifs that do not capture this signal, placing us in the conservative situation, described at the end of Section 4.7. The residual signal is not well explained by the mesh's centers, and thus its component on the orthogonal of the span of the activation vector of the second motif is important. The larger the mesh, the farther its center is to the selected motif and thus the less signal it captures, which explains the differences between the two curves.

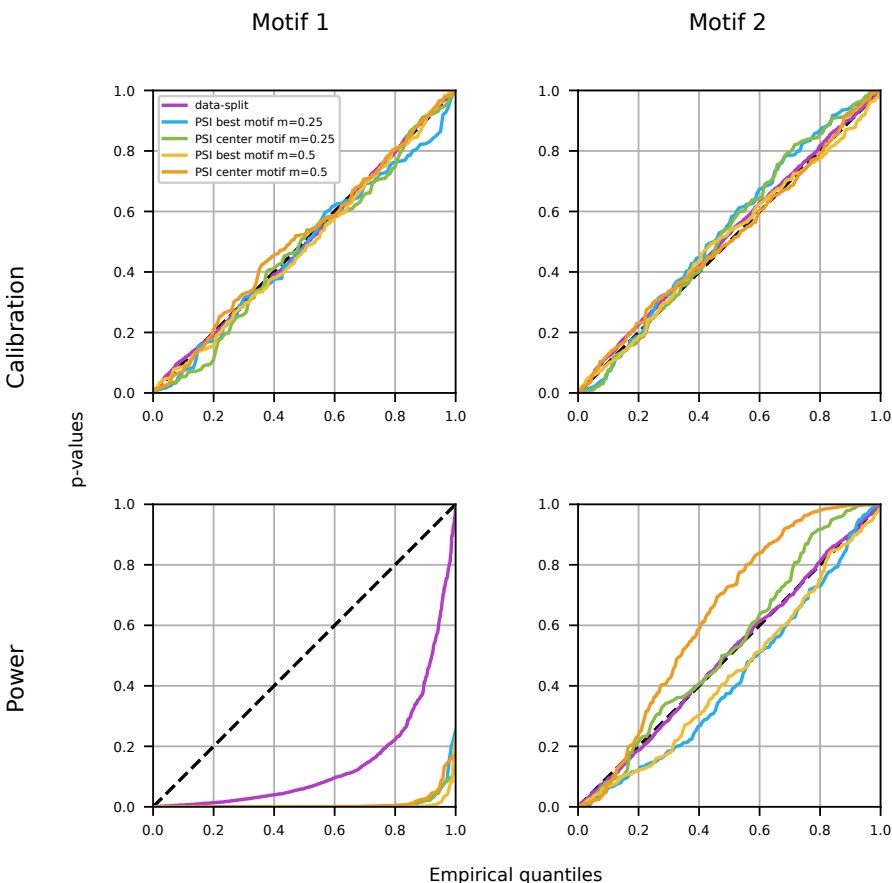

Figure 5: Q-Q plots obtained by applying data-split and different hit-and-run sampling strategies to select two motifs and test their association with an outcome. **Top**: data simulated under the null hypothesis. The proximity between the quantiles of the obtained p-values and those of the uniform distribution confirms that all SEISM strategies presented in this article are correctly calibrated. **Bottom**: data simulated under an alternative hypothesis, where the outcome depends on the activation $\varphi^{z,X}$ of a single motif in the sequence. The distributions of the p-values computed with the post-selection inference (PSI) strategies have a larger deviation to the uniform distribution than the distributions of the p-values computed with the data-split strategy (purple).

### 5.3 Computation costs

The section serves as an overview of how various user-specified parameters impact the computation time required by the post-selection inference procedure.

As discussed in 4.6, the hit-and-run algorithm is actually a rejection sampler. Its overall computation cost depends mainly on two characteristics: the cost of the selection step, that is the cost of selecting $q$ motifs for a given phenotype $\boldsymbol{y}$, and the acceptance rate. Although some parameters affect the selection cost, the acceptance rate is primarily responsible for determining if a user-specified combination of parameters results in a tractable configuration for the post-selection method in a reasonable amount of time. This rate is high compared with a naive rejection sampler over $\mathcal{E}$, as the hit-and-run strategy reduces the dimension over which the rejection step is performed: from $n$ with a naive sampler to 1. Nonetheless some parameters may have a major impact on the rejection rate. To clarify it, we studied in Figure 6 the impact of several user-specified parameters — the number of motifs to be discovered, the precision of the meshes, the regularization parameter of the ridge score and the number of computation cores allowed during the rejection step of the hit-and-run sampler.

Although the number of motifs to be found by SEISM undoubtedly affects the selection cost, we can roughly consider that this relationship is linear. The upper left figure in Figure 6, however, demonstrates that the influence on the overall computing cost is superlinear, in line with the exponential growth of the number of distinct selection events one may describe with a fixed mesh size. As a result, the post-selection process quickly becomes intractable for discovering and test more than a few motifs.

We make a similar observation for mesh precision: computation time grows exponentially with the number of bins used to define the meshing. This can be explained by the exponential relationship between the number of bins and the number of different meshes (and thus the rejection rate). Of note, mesh precision has no impact on the selection time, and therefore the computation time is entirely explained by the acceptance rate.

We observe that the greater the regularization parameter $\lambda$, the lower the computation time. This can be explained by detailing its impact on the rejection rate. To understand it, it is necessary to note that the motifs are not selected over $\mathcal{Z}$, but over a less constrained set as described in 3. They are only projected onto $\mathcal{Z}$ at the end of the whole procedure, to ease their interpretation. The meshes are then defined over a vectorial space, leading to an infinite number of meshes. Compared to a small regularization parameter, a higher $\lambda$ favors motifs resulting in a $\boldsymbol{\varphi}^{\boldsymbol{z},\boldsymbol{X}}$ with a higher norm. With regard to the activation function, such motifs are located closer from the $k$-mers, and thus from $\mathcal{Z}$. $\lambda$ has then no effect on the number of existing meshes, but impacts the number of *acceptable* ones, in the sense that they have a reasonable probability to be selected. A lower $\lambda$ leads to better selection performances, but to a higher number of acceptable meshes, and thus to a lower acceptance rate. We empirically set $\lambda = 0.01$ to provide a good trade-off.

Finally, the rejection sampling step can be parallelized over several computation cores, which accelerates the whole procedure, as described in Section 4.4. As long as the acceptance rate is small enough, using $j$ cores to parallelize the rejection step should roughly divide the computation time by $j$.

We can clearly identify limitations inherent to the use of the selective inference procedure. Although it is more powerful than the data-split approach, it can not be used in every situation. This latter approach does indeed not include any rejection step, and the only factor influencing its overall computation time is the selection time, only marginally influenced by the aforementioned parameters.

### 5.4 Using SEISM on empirical data

The experiments on simulated data that we present in Sections 5.2 and 5.3 are useful to analyse the calibration, power and computational behavior of SEISM in a controlled environment where the ground truth is known. Here we use an empirical dataset to evaluate two other critical aspects: the robustness of SEISM to the Gaussian assumption, and its ability to select the correct number of filters. We rely on the ChIP-seq dataset from Chatagnon et al. (2015). As part of this study, the authors investigate the mechanisms underlying cell differentiation and are particularly interested in the retinoic acid receptor (RAR), a transcription

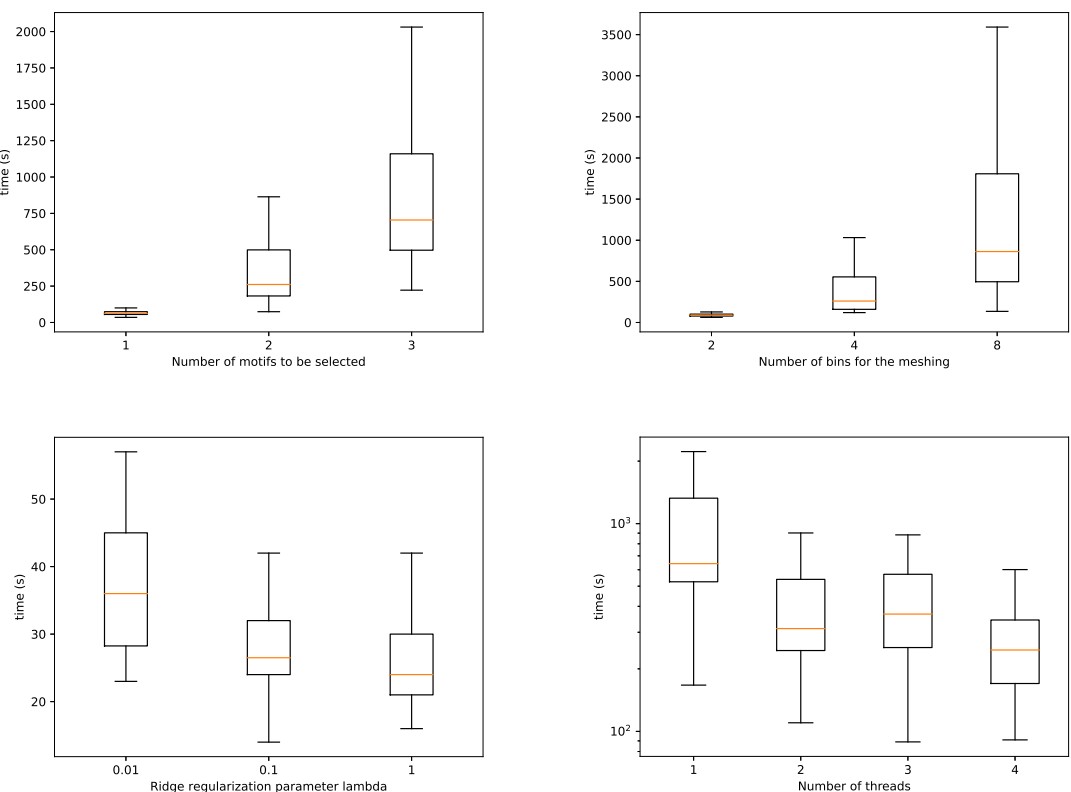

Figure 6: Impact of different parameters on the computation time for 100 replicates for the post-selection inference procedure. **Upper left:** Impact of the number of motifs to be discovered **Upper right:** Impact of the number of bins defining the meshes. **Bottom left:** Impact of the ridge regularization parameter. **Bottom right:** (Log scale) Impact of the number of threads over which the hit-and-run sampling is parallelized.

factor. They perform a ChIP-seq experiment, which leads to a dataset containing 131,895 sequences of length 500 and their associated phenotypes: the $-\log_{10}$ of $p$-values resulting from a test to determine whether the sequence is associated with a high number of bindings with RAR. We then derive a smaller dataset containing only 1,000 sequences, enriched with sequences significantly associated with RAR, in order to increase the signal to noise ratio and speed up the computations.

The Gaussian assumption is strongly challenged on classification labels, but can also be questioned for continuous phenotypes. Although this problem is not limited to SEISM, we propose here an approach to assess whether the Gaussian assumption is valid for a given dataset $(\boldsymbol{X}, \boldsymbol{y})$ for the SEISM procedure. This method follows 3 steps:

1. Create $N$ datasets $(\boldsymbol{X}, \boldsymbol{y}^{(i)})$ derived from the original one. The sequences are unchanged, but the labels are randomly permuted versions of $\boldsymbol{y}$. This permutation ensures that these new datasets are under the null hypothesis, while maintaining the original probability distribution of $\boldsymbol{y}$.

2. Run the whole SEISM procedure on each of those permuted datasets, and collect the $p$-values.

3. Draw a Q-Q plot: if the distribution is close to the uniform, then it validates the use of the Gaussian model for this dataset.

Our analysis in Section 5.3 suggests that the selective inference version of SEISM would be too computationally intensive and would bring little improvement compared to the data-split version on this dataset, and we therefore apply the above procedure with data-split. We use the ridge association score with a penlaty paramter $\lambda = 0.1$. The resulting empirical distribution of the labels is far from a Gaussian one (see Figure 7), but the $p$-values obtained on the permuted datasets with the aforementioned methodology are uniformly distributed between 0 and 1, as shown in Figure 8, which validates the use of SEISM on this particular dataset.

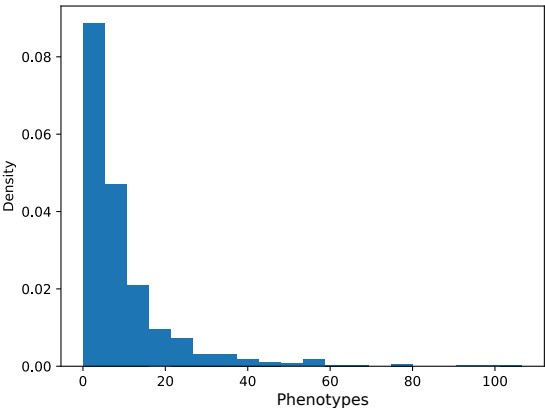

Figure 7: Empirical probability density of the phenotypes in Chatagnon et al. (2015).

We then select and test four motifs using the same data-split version of SEISM on the non-permuted dataset. The resulting motifs are represented in Table 2 with their respective $p$-values.

The first motif found— the one with the lowest $p$-value—recovers a known motif for RAR (Balmer & Blomhoff, 2005), see Figure 9. On the other hand, only the first discovered motif is associated with a significant $p$-value, aligning with the current literature for RAR. This confirms the capability of SEISM to infer a posterior the right number of feature in the model.

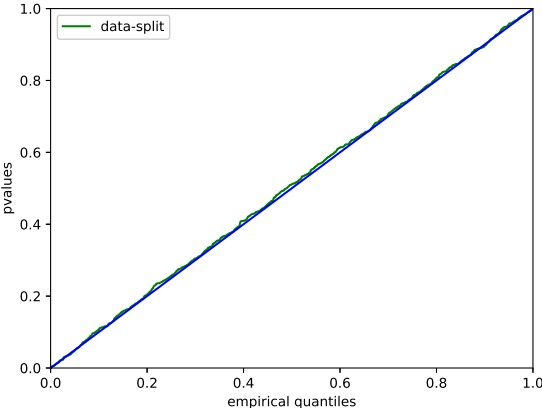

Figure 8: Q-Q plot obtained by applying the SEISM procedure to permuted versions of Chatagnon et al. (2015).

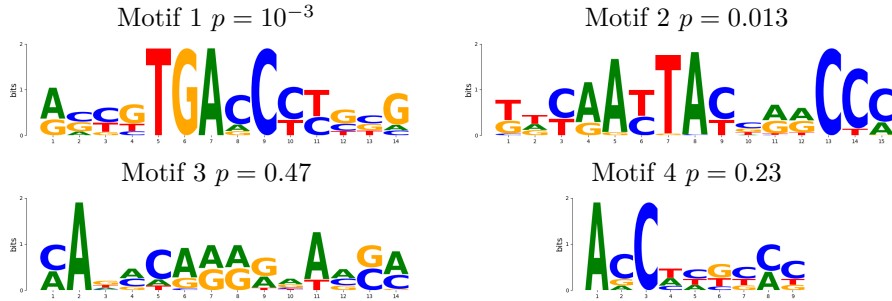

Table 2: Motifs and *p*-values obtained with SEISM on the dataset from Chatagnon et al. (2015).

## 6   Discussion and future works

We have introduced a procedure to test the association between features learned by a neural network and the outcome predicted by this network. We did so by relying on the post-selection inference framework and formalizing the network training as a feature selection step. Along the way, we addressed general problems related to selective inference over composite hypotheses, which has implications beyond testing of features extracted by trained neural networks. In particular to our knowledge, all previous procedures had to work under the assumption that the variance was known. Our strategy to normalize the statistic to make it scale-free could easily be transferred to kernelPSI for testing the association of kernels with a trait, or to previous selective inference frameworks for testing groups of variables using sampling strategies (Slim et al., 2019; Reid et al., 2018).

From a neural network perspective, SEISM provides a principled way to select the number of filters of a CNN, with a different objective—significance—than the usual prediction-oriented cross-validation pro-

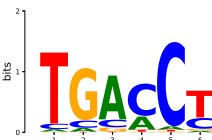

Figure 9: Known binding motif for RAR (Balmer & Blomhoff, 2005).

cedures. Through the SEISM procedure, we are also drawing connections between neural networks for biological sequences and two related fields: sequence motif detection, and GWAS.

Sequence motif detection has been a major theme in bioinformatics for the past 30 years and many methods have been proposed to identify motifs that are over-represented in a set of sequence compared to some control class or background distribution. The earliest CNNs for regulatory genomics Alipanahi et al. (2015); Zhou & Troyanskaya (2015) already exploited the fact that trained convolution filters of the first layer could be interpreted as PWMs, and more recent work have sought to extract PWMs from entire multi-layer trained networks through attribution methods. The selection step of our procedure merely formalizes that training a one-layer CNN is equivalent to selecting a finite set of PWMs that have a maximal association to the outcome for some particular score. This formalization also highlights the specific way by which CNNs with exponential activation functions parameterize the distribution of k-mers at a binding site. Although the PWM returned by most bioinformatics models represents a categorical distribution—probability to draw each letter at each site, trained convolution matrices parameterize a Gaussian distribution. In practice, this difference leads to discrepancies between the trained convolution filters and PWMs learned using categorical likelihoods—including those offered by databases and often used as ground truth. This observation also suggests alternative sets of constraints for convolution filters—*e.g.*, each column of the pointwise exponential of the filter should belong to the simplex.

By providing an inference procedure for features extracted by the trained model, our work also connects neural networks for genomic sequences to GWAS. The good predictive performances of these neural networks is often explained by their ability to jointly learn an appropriate data representation and a regular prediction function acting on this representation. Nonetheless, the space from which these representations are learned is seldom formalized and to our knowledge the association of the extracted features with the predicted outcome is never tested. GWAS on the other hand relies on hypothesis testing, but commonly relies on relatively simple genomic variants such as single nucleotide polymorphisms (SNPs) or $k$-mer presence (Jaillard et al., 2018; Roux de Bézieux et al., 2022). Our framework paves the way to GWAS over richer sets of variants, *e.g.* capturing the presence of entire polymorphic genes through large convolution filters, or the interaction of simpler variants through multilayer or self-attention networks (Avsec et al., 2021a). This will require scaling to entire genomes as inputs, and making more complex networks, such as multi-layer CNNs and networks using attention mechanisms, amenable to inference. The most important step in achieving this goal is to formulate the training of these networks as a feature selection problem and formalize the association between these features and the phenotype. The inference framework might then be directly derived from this present work. For instance, we may test motif interactions derived from convolutional-attention networks (Ullah & Ben-Hur, 2021), or a (motif, position) couple as selected in (Ditz et al., 2022). Regarding multi-layer CNNs, several strategies are conceivable. One solution would be to build on TFModISco, that aims at extracting motifs summarizing the features captured by a trained multi-layer CNN (in particular, accounting for possible interactions). These extracted motifs could be tested using the SEISM framework: the selection event is the set of simulated phenotypes that would lead to the construction of TFModISco motifs (Shrikumar et al., 2018) within the same meshes. Of note in this strategy, the motifs would not be selected using the same score used as a statistic for testing, but this is not an issue. A second, more integrated possibility would be to test the filters of the first layer within a deeper network. Deeper layers indeed model interactions between the motifs, and with such an architecture the filters of the first layer would then be optimized while taking into account those interactions. The selection procedure would still be a greedy one, and the architecture of the network would vary from step to step: the first layer would contain more and more filters, but the previously entered filters would be fixed. This selection procedure could then be translated into a selection event, and the inference framework could then be applied accordingly. Finally, we could test the deeper features themselves, but this would only make sense for an appropriate architecture that makes these features interpretable. A simple option would be, for example, to apply a global (or large enough) pooling over a first layer with few filters, and test filters of the second layer that would represent motif combinations. Alternatively, the second layer could be an even simpler set of pairwise interactions between motifs (*i.e.*, special filters with only two non-zero entries). In practice, this could be done by optimizing the residual errors for the successive filters of this second layer. Admittedly, a few practical problems may arise. First, the hit-and-run sampler requires the selection method to be stable, that is, running the selection method twice on the same input will lead to the same selection on features. This property is required to guarantee the

theoretical convergence of the the algorithm but may not be necessary in practice. Second, some attention may be required to avoid the that computational cost become prohibitive, in particular depending on the regularity properties of the selection event leading to a higher rejection probability or to a higher number of replicates required. Granted that these technical challenges can be addressed, we are confident that extending SEISM to more general networks and corresponding features will benefit both the fields currently using these networks—such as regulatory genomics—and GWAS.

## 7 Acknowledgements

This work has been supported by ANR grants (FAST-BIG project ANR-17-CE23-0011-01 and PIECES project ANR-20-CE45-0017) and was performed using the computation facilities of the LBBE/PRABI.

We thank François Gindraud, Jean-Philippe Rasigade, Lotfi Slim and Dexiong Chen for the insightful discussions and support.

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

# Supplemental Materials of "Neural Networks beyond explainability: Selective inference for sequence motifs"

## A    Tuning the activation bandwidth hyperparameter

The data representation $\boldsymbol{\varphi}^{\boldsymbol{Z},\boldsymbol{X}}$ depends on a hyperparameter $\omega$ controlling the bandwidth of the gaussian non-linearity (Equation 3): $\exp\left(-\frac{||\boldsymbol{z}_j - \boldsymbol{u}||^2}{2\omega^2}\right)$. Assuming that the positions are independant, we know that the expected value of the distance between a motif $\boldsymbol{z}$ and a $k$-mer $u$ with length $k$ is proportional to $k$.

In order to get an activation that does not depend on the length of the motifs, we simply set $\omega$ to be proportional to $\sqrt{k}$. From empirical tests, we set $\omega = \frac{\sqrt{0.9 * k}}{2}$ to achieve good selection results by choosing the motif that maximizes the association score among a set of possible lengths.

## B    Disintegration of the selection event given by sequence motifs

In this section we consider the selection event:

$$E_{\text{cont.}}(\boldsymbol{Z}) := \left\{\boldsymbol{y}' \in \mathcal{E}, \ \forall i \in \{1, \ldots, q\} \ \underset{\boldsymbol{z} \in \mathcal{Z}}{\arg\max}\, s(\boldsymbol{z}, \boldsymbol{P}_i \boldsymbol{y}') = \boldsymbol{z}_i\right\}, \tag{S1}$$

given by the sequence of selected motifs $\boldsymbol{Z} = (\boldsymbol{z}_1, \ldots, \boldsymbol{z}_q)$. We denote by $\mu$ the law of $\boldsymbol{y}$ as given by Eq. (9), a Gaussian distribution on $\mathcal{E}$.

### A first remark on the uniqueness of the selection

Consider the mapping $\pi : \mathcal{E} \to \mathcal{Z}^q$ given by $\pi(\boldsymbol{y}') = \boldsymbol{Z}$ where $\boldsymbol{Z} = (\boldsymbol{z}_1, \ldots, \boldsymbol{z}_q)$ is the sequence of motifs such that $\boldsymbol{y}' \in E_{\text{cont.}}(\boldsymbol{Z})$. It is not clear that $\pi$ is well defined as a same $\boldsymbol{y}'$ may lead to the selection of at least two different motifs sequences $\boldsymbol{Z}$ and $\boldsymbol{Z}'$. As a first remark, we can see that the set of problematic $\boldsymbol{y}'$ is exactly

$$\mathcal{P} := \bigcup_{\boldsymbol{Z} \neq \boldsymbol{Z}'} E_{\text{cont.}}(\boldsymbol{Z}) \cap E_{\text{cont.}}(\boldsymbol{Z}').$$

When one assumes that $\boldsymbol{Z} = (\boldsymbol{z}_1, \ldots, \boldsymbol{z}_q)$ is unique, one implicitly assumes that $\mu(\mathcal{P}) = 0$. For sufficiently regular scores, this is however the case. For sake of readability, we will not comprehensively study this issue but we will present an argument for the scores $s^{\text{HSIC}}$ and $s^{\text{ridge}}$. In this case, we can circumvent this difficulty considering the Gaussian random field

$$\boldsymbol{z} \mapsto \langle \boldsymbol{\varphi}^{\boldsymbol{z},\boldsymbol{X}}, \boldsymbol{y} \rangle \text{ for (HSIC)} \quad \text{and} \quad \boldsymbol{z} \mapsto \langle \left(\|\boldsymbol{\varphi}^{\boldsymbol{z},\boldsymbol{X}}\|^2 + \lambda n\right)^{-1/2} \boldsymbol{\varphi}^{\boldsymbol{z},\boldsymbol{X}}, \boldsymbol{y} \rangle \text{ for (Ridge)}$$

indexed by $\mathcal{Z}$ where $\boldsymbol{y}$ is distributed with respect to a multivariate Gaussian distribution Eq. (9). Its autocovariance function is given by $(\boldsymbol{z}, \boldsymbol{z}') \mapsto \sigma^2 \langle \boldsymbol{\varphi}^{\boldsymbol{z},\boldsymbol{X}}, \boldsymbol{\varphi}^{\boldsymbol{z}',\boldsymbol{X}} \rangle$ from Eq. (9) (one has to multiply by $\left(\|\boldsymbol{\varphi}^{\boldsymbol{z},\boldsymbol{X}}\|^2 + \lambda n\right)^{-1/2}\left(\|\boldsymbol{\varphi}^{\boldsymbol{z}',\boldsymbol{X}}\|^2 + \lambda n\right)^{-1/2}$ for the Ridge). The score is just the largest norm of this Gaussian random field. It is well established in theory of Gaussian random fields that the law of this maximum is regular and the argument maximum is unique. The interested reader may consult the pioneering work of Tsirelson (Tsirelson, 1976) and Lifshits (Lifshits, 1983). In Tsirelson's theorem, the parameter set is countable. This says that the same result holds true for separable bounded Gaussian processes, since in this case, the distribution of the supremum coincides a.s. with the one of the supremum on some countable nonrandom set. To avoid a cumbersome presentation, we will assume that almost surely the selected sequence motifs $\boldsymbol{Z} = (\boldsymbol{z}_1, \ldots, \boldsymbol{z}_q)$ is uniquely defined, hence $\pi$ is well defined.

**The disintegration steps**

To sample conditionally on (S1), one need to consider the conditional law with respect to this event. We will denote this law by $\mu_{\boldsymbol{Z}}$, it depends only on $\mu$, $\boldsymbol{Z}$ and $\pi$. This law is described by the theorem of disintegration, see for instance (Ambrosio et al., 2005, Theorem 5.3.1). Denote $\nu$ the pushforward measure of $\mu$ by $\pi$, denoted by $\nu = \pi_\# \mu$, a probability measure on the set $\mathcal{Z}^q$ of $\boldsymbol{Z}$. By the disintegration theorem, there exists a $\nu$-almost everywhere uniquely determined Borel family of probability measures $\mu_{\boldsymbol{Z}}$ (the though-after conditional distributions) such that

- **Supported by $E_{\text{cont.}}(\boldsymbol{Z})$:** $\mu_{\boldsymbol{Z}}\{\mathcal{E} \setminus \pi^{-1}(\boldsymbol{Z})\} = 0$ for $\nu$-almost every $\boldsymbol{Z}$;

- **Expectation of the conditional expectation is the expectation:** It holds that, for every Borel test map $f : \mathcal{E} \to [0, +\infty]$,

$$\int_{\mathcal{E}} f \mathrm{d}\mu = \int_{\mathcal{Z}^q} \Big( \int_{\pi^{-1}(\boldsymbol{Z})} f \mathrm{d}\mu_{\boldsymbol{Z}} \Big) \mathrm{d}\nu(\boldsymbol{Z}), \tag{S2}$$

where one can remark that $\pi^{-1}(\boldsymbol{Z}) = E_{\text{cont.}}(\boldsymbol{Z})$ by definition of $\pi$. Let us comment on this result regarding our purposes. First, we have mentioned that we known that the support $E_{\text{cont.}}(\boldsymbol{Z})$ is included in some subspace, say $\mathcal{S}$, defined by the first order conditions. Second, although one can use a rejection sampling strategy on the subspace $\mathcal{S}$ to draw points on the support $E_{\text{cont.}}(\boldsymbol{Z})$ (viewed as a subset of the same Hausdorff dimension as the subspace $\mathcal{S}$), it is not clear at all what should be the density of $\mu_{\boldsymbol{Z}}$. Indeed, the family of probability measures $\mu_{\boldsymbol{Z}}$ is the unique family that satisfies Eq. (S2). It implies that a measure $\mu_{\boldsymbol{Z}}$ depends on the others measures $\mu_{\boldsymbol{Z}'}$ and this dependency is geometrically given by the (piece-wise) topological sub-manifold given by the function $\boldsymbol{z} \mapsto \boldsymbol{\varphi}^{\boldsymbol{z}, \boldsymbol{X}}$ from $\mathcal{Z}$ to $\mathcal{E}$.

From a practical view point, we tried various law for $\mu_{\boldsymbol{Z}}$ such as the uniform, or a rejection sampling based on the Gaussian distribution (9), but none of them matched the condition (S2). In the next subsection, we recall a toy example: the disintegration of the uniform measure on the sphere is not the uniform measure. Even in this simple geometrical example, the calculus of the conditional law might be seen as tedious. We believe that the calculus of $\mu_{\boldsymbol{Z}}$ is somehow out of reach for our purposes and our analysis with selection events defined by meshes more suited.

**A toy example on the sphere**

Let \$ be the 2-sphere embedded in the 3-Euclidean space. Let $\mu$ be the uniform measure on the sphere \$. Let $\{\mathcal{S}_\theta : \theta \in [0, \pi)\}$ be a family of sub-spaces of co-dimension 1 (hyper-planes) sharing $\text{Span}\{(0, 0, 1)\}$ (say the north pole) as a revolution axis parameterized by $\theta$. The parameter $\theta$ can be interpreted as the longitude.

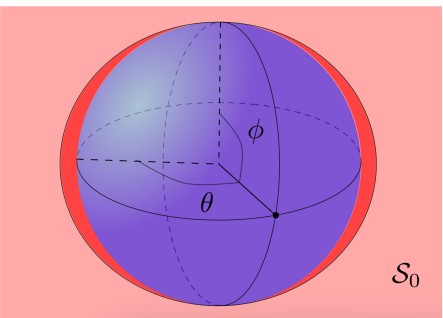

Figure S1: For $\theta = 0$, $S_\theta$ is the light red plan, the conditional measure $\mathrm{d}\mu_0(\phi)$ is depicted with a red area and is proportional to $|\sin(\phi)|$, which is not the uniform measure.

Let $\bar{\pi}$ be the function that maps a point to its longitude modulo $\pi$. By spherical symmetries, the pushforward measure $\nu = \bar{\pi}_{\#}\mu$ is the uniform measure on $[0, \pi)$, so that $\mathrm{d}\nu(\theta) = (1/\pi)\mathrm{d}\theta$. Condition (S2) (the lhs of the equality below) is given by the coordinate integration system (the rhs) in:

$$\int_{\mathbb{S}} f\mathrm{d}\mu = \int_0^{\pi}\Big(\int_{\bar{\pi}^{-1}(\theta)} f\mathrm{d}\mu_{\theta}\Big)\mathrm{d}\nu(\theta) = \int_0^{\pi}\Big(\int_0^{2\pi} f(\theta, \phi)\frac{|\sin \phi|}{4\pi}\mathrm{d}\phi\Big)\mathrm{d}\theta\,,$$

where $\phi$ is the latitude. Note that $\bar{\pi}^{-1}(\theta) = \mathbb{S}\cap\mathcal{S}_{\theta}$ and it is in bijection with $[0, 2\pi)$ using the mapping that to a point maps its latitude. Using this representation, it is not hard to see that the uniform measure on $\bar{\pi}^{-1}(\theta)$ is given by $(1/2\pi)\mathbf{1}_{[0,2\pi)}(\phi)$ while the above equality shows that the conditional measure $\mu_{\theta}$ of the uniform measure on the sphere has density $(1/4)|\sin \phi|\mathbf{1}_{[0,2\pi)}(\phi)$, see Figure S1. It proves that the disintegration of the uniform measure on the sphere is not the uniform measure, but rather a distribution that will put few mass around the poles and large mass around the equator.

## C Proof of Proposition 4.4

Consider the orthogonal decompositon

$$\mathcal{E} = \mathcal{R} \oplus S \oplus \mathcal{T}$$

where $\mathcal{R}$ is the span of $\boldsymbol{\varphi}^{\boldsymbol{Z},\boldsymbol{X}}$, $\mathcal{T}$ is the span of $\boldsymbol{\mu}$ (orthogonal to $\mathcal{R}$ by Proposition 4.1), and $\mathcal{S}$ such that the equality holds. Consider $\boldsymbol{y} \in \mathcal{E}$ and its othogonal decomposition $\boldsymbol{y} = \boldsymbol{r} + \boldsymbol{s} + t\boldsymbol{e}$ where $\boldsymbol{e} = \boldsymbol{\mu}/\|\boldsymbol{\mu}\|_2$ is a unit norm vector that spans $\mathcal{T}$. Let $\tau > 0$ and note that it is enough to prove that

$$\mathbb{P}_{\boldsymbol{\mu}}\Big\{\frac{s(\boldsymbol{Z},\boldsymbol{Y})}{\|\boldsymbol{Y}\|^2} \leq \tau\Big\} \geq \mathbb{P}_{\boldsymbol{0}}\Big\{\frac{s(\boldsymbol{Z},\boldsymbol{Y})}{\|\boldsymbol{Y}\|^2} \leq \tau\Big\}\,,$$

where $\boldsymbol{Y}$ is a random variable with the same distribution as $\boldsymbol{\mu} + \sigma\boldsymbol{\epsilon}$ (resp. $\sigma\boldsymbol{\epsilon}$) on the probability space defined by $\mathbb{P}_{\boldsymbol{\mu}}$ (resp. $\mathbb{P}_{\boldsymbol{0}}$). Not that the event decomposed as

$$\Big\{\boldsymbol{y} \,:\, \frac{s(\boldsymbol{Z},\boldsymbol{y})}{\|\boldsymbol{y}\|^2} \leq \tau\Big\} = \Big\{(t, \boldsymbol{r}, \boldsymbol{s}) \,:\, s(\boldsymbol{Z},\boldsymbol{r}) \leq \tau(t^2\|\boldsymbol{\mu}\|^2 + \|\boldsymbol{r}\|^2 + \|\boldsymbol{s}\|^2)\Big\}$$

By othogonality, note that $\mathcal{L}_{\boldsymbol{\mu}}(\boldsymbol{r}, \boldsymbol{s}) = \mathcal{L}_0(\boldsymbol{r}, \boldsymbol{s})$ and this law is a centered Gaussian multivariate law. We deduce that the aforementioned probabilities are of the form

$$\mathbb{P}_{\boldsymbol{\mu}}\Big\{\frac{s(\boldsymbol{Z},Y)}{\|Y\|^2} \leq \tau\Big\} = \int_0^{\infty} w_0(t)\varphi_{\mu}(t)\mathrm{d}t$$

where

$$w_0(t) = \mathbb{P}_0\Big\{s(\boldsymbol{Z},\boldsymbol{r}) \leq \tau(t^2\|\boldsymbol{\mu}\|^2 + \|\boldsymbol{r}\|^2 + \|\boldsymbol{s}\|^2)\Big\}$$
$$\varphi_{\mu}(t) = \exp\big(-(t - \mu_e)^2/2\big) + \exp\big(-(t + \mu_e)^2/2\big)$$

with $\mu_{\boldsymbol{e}} = \langle \boldsymbol{e}, \boldsymbol{\mu}\rangle = \|\boldsymbol{\mu}\|_2$. Note that $w_0 : (0, \infty) \to (0, 1)$ is an increasing continuous function. It is an increasing homeomorphism and the Fubini's equality yields

$$\begin{aligned}
\mathbb{P}_{\boldsymbol{\mu}}\Big\{\frac{s(\boldsymbol{Z},Y)}{\|Y\|^2} \leq \tau\Big\} &= \int_0^{\infty} w_0(t)\varphi_{\mu}(t)\mathrm{d}t \\
&= \int_0^{\infty}\int_0^1 \mathbf{1}_{\{u \leq w_0(t)\}}\mathrm{d}u\,\varphi_{\mu}(t)\mathrm{d}t \\
&= \int_0^{\infty}\int_0^1 \mathbf{1}_{\{w_0^{-1}(u) \leq t\}}\mathrm{d}u\,\varphi_{\mu}(t)\mathrm{d}t \\
&= \int_0^1\int_{w_0^{-1}(u)}^{\infty} \varphi_{\mu}(t)\mathrm{d}t\mathrm{d}u
\end{aligned}$$

By Anderson's theorem, the measure of the interval $[-w_0^{-1}(u), w_0^{-1}(u)]$ for the centered Gaussian density is greater than the one for a non-centered Gaussian density with the same variance. As a result, we deduce that

$$\int_{w_0^{-1}(u)}^{\infty} \varphi_\mu(t)\mathrm{d}t \geq \int_{w_0^{-1}(u)}^{\infty} \varphi_0(t)\mathrm{d}t\,,$$

which achieves the proof.

