# OpenReview forum: "Neural Networks beyond explainability: Selective inference for sequence motifs"
_TMLR — Accepted by TMLR_

### Review · Reviewer_FHuH · 2023-01-11

**Summary Of Contributions:**

The paper describes an interpretable method called SEISM for detecting and testing the associations between sequence motif features learned by a convolutional neural network and the outcome phenotypes predicted by the CNN. The method uses a one-layer CNN as an interpretable sequence motif discovery tool, and leverages a post-selection inference framework to select over a continuous set of features. The motif discovery performance of SEISM is benchmarked against de novo motif discovery baselines (STREME, MEME, CKN-seq), and shows comparable performance.

**Audience:**

Yes

**Claims And Evidence:**

Yes

**Requested Changes:**

Some minor comments:
Figure 4 right, the error bars for the different methods are overlapping and hard to read


**Strengths And Weaknesses:**

Strengths:

A code implementation is provided

Proposed method performs comparably to other baselines in the motif discovery benchmark

Other questions/comments:

“For the sake of simplicity, we choose to restrict this presentation to simple one-layer CNNs and sequence motifs. The procedure that we introduce, however, is by no means limited to this framework, and could be applied to any of the more expressive features proposed in the explainable machine learning literature.” -- But the authors mention the one-layer CNNs are a special straightforward case for providing interpretable results since the trained filters can be seen as position weight matrices. What about multilayer CNNs where interpretability is more challenging?

“The user must also specify the number of motifs to find, as well as a parameter controlling the meshing of the motif space, that is the precision with which the found motifs will be tested.” -- How sensitive are the results to the choice of these hyperparameters? How should the number of motifs be determined a priori?

“To that end, we enumerates the k-mers contained in X using the DSK software (Rizk et al., 2013) and compute their scores s(·, ·)” -- How is k set?

“In order to assess the statistical validity and of the SEISM procedure with the different strategies, we simulate datasets under the null hypothesis. To that end, we draw one sequence motif z˜ with length k = 8 for each simulated dataset using a uniform distribution on Z restricted to motifs with an information level fixed at 10 bits” -- Why only analyze a sequence with length k = 8?

It is unclear to me what are the advantage/disadvantage of the “de novo” motif discovery methods (STREME, MEME) compared to the proposed SEISM method (their performances in Figure 4 are very similar)

---

> ### Author Response · Authors · 2023-04-23
> **To Reviewer FHuH (1/2)**
>
> Thank you for your insights and your valuable comments. Here is a summary of our answers to your questions/comments:
>
>     • "For the sake of simplicity, we choose to restrict this presentation to simple one-layer CNNs and sequence motifs. The procedure that we introduce, however, is by no means limited to this framework, and could be applied to any of the more expressive features proposed in the explainable machine learning literature.” – But the authors mention the one-layer CNNs are a special straightforward case for providing interpretable results since the trained filters can be seen as position weight matrices. What about multilayer CNNs where interpretability is more challenging?
>
> → We can identify three strategies to perform inference over features extracted from multilayer CNNs:
>
> One solution would be to build on TFModISco, that aims at extracting
> motifs summarizing the features captured by a trained multi-layer CNN
> (in particular, accounting for possible interactions). These extracted
> motifs could be tested using the SEISM framework: the selection event
> is the set of simulated phenotypes that would lead to the construction
> of TFModISco motifs within the same meshes. Of note in this strategy,
> the motifs would not be selected using the same score used as a
> statistic for testing, but this is not an issue.
>
> A second, more integrated possibility is to test the filters of the
> first layer within a deeper network. Deeper layers indeed model
> interactions between the motifs, and with such an architecture the
> filters of the first layer would then be optimized while taking into
> account those interactions.  The selection procedure would still be a
> greedy one, and the architecture of the network would vary from step
> to step: the first layer contains more and more filters, but the
> previously entered filters are fixed. This selection procedure can
> then be translated into a selection event, and the inference framework
> can then be applied accordingly.
>
> Finally, we could test the deeper features themselves, but this would
> only make sense for an appropriate architecture that makes these
> features interpretable. A simple option would be, for example, to
> apply a global (or large enough) pooling over a first layer with few
> filters, and test filters of the second layer that would represent
> motif combinations. Alternatively, the second layer could be an even
> simpler set of pairwise interactions between motifs (ie, special
> filters with only two non-zero entries). In practice, this could be
> done by optimizing the residual errors for the successive filters of
> this second layer.
>
> In order to clarify this question, we inserted the above discussion in the ’Discussion and future works’ section of our manuscript.
>
> We also modified the aforementioned sentence in the introduction, using: "In this paper, we restrict this presentation to simple one-layer CNNs and sequence motifs. The procedure we introduce here, however, is not limited to this framework. It can be applied to more expressive features proposed in the explainable machine learning literature, but may require some further work depending on the feature considered".
>
> We thank the reviewer for giving us the opportunity to clarify this point
>
> (Continued in the following answer)

---

> > ### Author Response · Authors · 2023-04-23
> > **To Reviewer FHUM (2/2)**
> >
> >  • "The user must also specify the number of motifs to find, as well as a parameter controlling the meshing of the motif space, that is the precision with which the found motifs will be tested.” – How sensitive are the results to the choice of these hyperparameters? How should the number of motifs be determined a priori?
> >
> > → The size of the meshes has implications on the computation time, but also on the statistical power. As the meshes become finer, the computation time increases while the power decreases (Optimal Inference After Model Selection, Fithian et. al 2014, Proposition 3). In our experience, working with fairly large meshes (with only two or three bins) led to satisfying results. The size of the mesh relates to the precision of the null hypothesis, and eventually depends on the needs of the user.
> >
> > The number of motifs to be found can be fixed according to different strategies:
> >
> > -It can be fixed a priori depending on the knowledge on the data. For instance, a transcription factor usually has no more than two to three motifs.
> >
> > -During the selection procedure, we can decide to stop the greedy procedure at a given step j if the score of the new motif is below a given threshold, which may possibly depend on the scores of the previously entered filters, similar to an elbow rule.
> >
> > -Finally, we want to emphasize that this issue is classical in machine learning: the architecture of a CNN is fixed beforehand and is one of the hyper parameters of the model. SEISM offers a way to adjust this parameter a posteriori, by removing from the model the filters that are not significant.
> >
> > We added the following sentence to our Discussion section to highlight this point:
> >
> > “From a neural network perspective, SEISM provides a principled way to select the number of filters of a CNN, with a different objective---significance---than the usual prediction-oriented cross-validation procedures.”
> >
> >     • « To that end, we enumerates the k-mers contained in X using the DSK software (Rizk et al., 2013) and compute their scores s(·, ·) ». How is k set?
> >       « In order to assess the statistical validity of the SEISM procedure with the different strategies, we simulate datasets under the null hypothesis. To that end, we draw one sequence motif z with length k = 8 for each simulated dataset using a uniform distribution on Z restricted to motifs with an information level fixed at 10 bits ». Why only analyze a sequence with length k = 8?
> >
> > When SEISM is applied to real datasets, k is first set across a range according to the user, and then the optimal k is found by SEISM, as described in Annex A. We modified section 3.2 to clarify this point: “The k-mer list is obtained using the DSK software (Rizk et al., 2013). The length k first varies according to a user-defined range, and the optimal value is chosen by SEISM, as described in Appendix A.”
> >
> >
> > In the simulation, we used a fixed value k=8 to reduce the computation time. Any other value for k would also lead to calibrated results, but a bad choice for k may indeed result in a loss of statistical power. Moreover, in this experiment we are interested in the performance gain allowed by selective inference compared to data split, and we used the same value k=8 for the data split split strategy.
> >
> >     • It is unclear to me what are the advantage/disadvantage of the “de novo” motif discovery methods (STREME, MEME) compared to the proposed SEISM method (their performances in Figure 4 are very similar)
> >
> > SEISM is a general framework that may be used to select and test a richer set of features than the ones selected with "de novo" motif discovery tools. It is applied to this particular existing problem to show that it performs as well as state-of-the-art methods.
> > In "de novo'' motif discovery tasks, the main advantage of SEISM is the inference step on small datasets, SEISM uses the complete set of sequences for both selection and test, whereas MEME/STREME rely on data-split strategies.
> > Therefore, we hope to gain performance in selection and power. In the results provided in Figure 4, MEME and STREME can not test the discovered motifs, as they already used the entire dataset for the selection, while SEISM can.
> > SEISM is a very general method, as the assumptions on the association score are not stringent (lots of frameworks, such as linear regression models, fit this framework), which can show its full potential where data-split strategies are not usable.

---

### Review · Reviewer_rRMK · 2023-02-14

**Summary Of Contributions:**


This work proposes SEISM, a framework for extracting features/motifs from biological sequences and (statistically) associating them with target phenotypes. For feature extraction, the framework employs a single-layer (convolutional) neural network with radial basis activation and max pooling. The framework furthermore proposes a hypothesis testing procedure to establish (the statistical significance of) associations between the extracted features (z_j) and prediction targets/phenotypes y.

The authors define an association score as their optimisation objective, which is a function of feature and prediction targets, and it is derived from the solution of linear ridge regression. They propose an iterative greedy approach for learning features such that in every iteration, a new feature is learnt to model the residuals targets from previously learnt features. The size (k) and number of features or motifs (q) are hyper-parameters of the the feature extraction module.

Once the features are learnt, the authors propose a post-selection inference step to estimate p-values for establishing the statistical significance of association between each feature and target vector y. To that end, authors propose hypothesis testing which assumes a null hypothesis that is defined to have zero association score between a motif and the vector of target outcomes, implying orthogonality between the two entities. To estimate p-values for hypothesis testing, the authors propose to empirically compute the null distribution of association scores. They partition the feature space into a mesh structure so that the feature space spanned by the mesh partitions of extracted motifs can be sampled for the estimation of the null distribution of association scores. To simulate alternative target outcomes y_prime which lead to the selection of other motifs from the mesh partitions of learnt motifs, the authors propose a hit-and-run MCMC sampler. For computing the null distribution and corresponding test statistic, the authors put forth two definitions of null hypothesis based on two varying representations of mesh partitions. They also present theoretical analysis on the approximation of the null distribution through sampling from a global distribution.

In numerical analysis, SEISM is benchmarked against three other methodologies on a motif discovery task. The authors use a publicly available set of datasets to compare the accuracies/alignment of motifs discovered by different methodologies with the known ground-truth. They compute so called Tomtom score as a measure of accuracy. They apply thresholds to the best Tomtom scores achieved for each dataset by each of the compared methods to compute their success rates across all datasets. The authors find that for low threshold levels, their proposed method is comparable or better than the other approaches, whereas for higher levels of threshold, two of the compared methodologies tend to achieve better success rates in general.

Next the authors present calibration experiments using synthetic data, where they show that SEISM is on-par with data-split strategy for learning and inference in terms of empirically recovering the underlying null distribution of p-values. In further experiments on small synthetic datasets where the outcomes were generated under alternative hypothesis, SEISM is shown to be more salient in deviating from the null distribution than data-split strategy.

Last set of experiments demonstrate non-linear/exponential scaling behaviour of SEISM with respect to various hyper-parameters.

**Audience:**

Yes

**Broader Impact Concerns:**

No. concerns in this regard

**Claims And Evidence:**

No

**Requested Changes:**

Critical:

I would like to understand why to define a selection region for pose-selection inference, the authors propose to partition the feature space into a rigid mesh structure instead of for instance defining the selection regions to be within a certain distance from extracted features. Wouldn’t that lead to more uniform sampling from the space surrounding the extracted features?

It seems that in the greedy optimisation proposed by the authors, the order in which features are learnt would influence their relevance to the outcome. Can the authors show if there is in general any correlation between the order of extracted features and their associated p-values? In case that the order of features is found to have an influence on the association between features and outcome, would it not undermine the relevance of (computationally costly) post-selection inference procedure for assessing the associations between individual features and outcome?

To substantiate the claim that the proposed framework is generic and flexible, the authors should perform numerical experiments on different types of data. They should also demonstrate the scalability of their framework beyond the small scale experiments (2 or 5 features with coarse meshing) in the paper.

As post-selection inference is the main contribution of the paper, it feels imperative that the method is evaluated for its performance on extraction/selection of the “right” number of latent features from observed data. This can be demonstrated through (generated) data with known ground-truth.

Nice to have:

Through experiments, authors can also show how post-selection inference works for features learnt by different methods (for different input and targets). They can perform experiments with varying CNNs

It would be nice to see experiments that show how the bin size affects the performance of post-selection inference, especially when the input cardinality, size and/or number of features increase.

For feature selection/importance assessment, it would be nice to contrast the proposed method with existing approaches e.g., the Shapley Value.


**Strengths And Weaknesses:**

Strengths:

The work proposes a holistic framework for extraction and association of features with learning targets.

The proposed approach for post-selection inference can deal with continuous feature space

The authors provide theoretically guided parameterisation of null distribution, which enables sampling for post-selection inference procedure. They show that under the assumed null hypotheses and the Gaussian outcome model, the parameterisation of the distribution is invariant to scaling and translation.

The proposed method achieves competitive performance on motif discovery tasks. Experiments on synthetic data validate the proposed inference procedure

The framework may outperform data-split ]earning and inference strategy in small data regime

Weaknesses:

The major weakness of the proposed approach is that it is not scalable. Parameters such as the size of the input alphabet, size and number of features, granularity of the mesh structure each cause super-linear/exponential increase in the computational cost of the inference procedure. Even the experiments in the paper are rather small scale. It is therefore not obvious if the proposed framework is broadly applicable on a real scale compared to what the experiments demonstrate in the paper.

Although post-selection inference is the main contribution of this paper, the work falls short of demonstrating its utility in associating/inferring the right number of features, which bears significance for any real application

The framework assumes Gaussian noise model for targets, but it is not clear if it in general is a valid assumption for real application scenarios of the framework and what could be the implications when the assumption does not hold. Furthermore, the Gaussian activation units used in the convolutional neural network do not make an obvious choice for modelling one-hot encoded categorical input.

The framework is proposed to be generic and flexible, but it is only applied to one type of data. In regards to flexibility no results or variations in the components of the framework have been reported.

---

> ### Author Response · Authors · 2023-04-23
> **To Reviewer rRMK (1/2)**
>
> We thank the reviewer for the thoroughness of their feedback and insightful comments.
>
> Several concerns are raised about the scalability of SEISM. Depending on the parameter at hand, we can follow different strategies:
>
> -The most critical concern is admittedly the scalability in the number of sequences. We acknowledge that our selective inference approach is not scalable, has little practical use on the real world data that we present in our manuscript, and mostly serves as a conceptual progress. We believe that this contribution is nonetheless valuable, and could inspire future works that would improve its scalability. Importantly, as detailed in our comment to all reviewers on the top of our rebuttal, we also want to highlight that the scalable (albeit less powerful) data-split strategy is also novel in this context, and crucially depends on the theoretical results features in Propositions 4.2, 4.3 and 4.4.
>
> -Regarding the size of the meshes, at first glance it might seem to be a good idea to test very small meshes: the size of the mesh indeed relates to the resolution of the null hypothesis. However, Fithian et. al 2014 (Proposition 3) suggests that as the meshes become finer, the statistical power decreases (in addition to becoming computationally intractable). Working with fairly large meshes (with two or three bins) then provides satisfying results in our experience (i.e. the statistical power remains much better than the one obtained with data-split, while keeping a sufficient precision on the motifs), and remains acceptable in terms of computation time.
>
> -The number of motifs also impacts the computation time, as discussed in the results section. For some biological problems, such as the characterization of the binding motifs associated with transcription factors, this is not a big issue as such TFs only have a very limited number of binding motifs (less than 3 or 4). For other features, the more scalable data-split strategy is always possible.
>
> Regarding both the ability to infer the right number of features and the robustness to the Gaussian assumption, we added a new experiment on a real dataset in the ’Results’ section. In this experiment, based on a Chip-seq dataset to detect the binding motifs associated with the Retinoic Acid Receptor, we first give a fast method (and apply it) to check that the p-values given by SEISM on this particular dataset are valid, even if the labels are not distributed according to a Gaussian law: in a few words, this method consists in generating permuted datasets under the null hypothesis (with the same sequences as the original one and with the same label distribution). If SEISM gives calibrated p-values, it means that it is robust to the Gaussian hypothesis for this dataset. Moreover, we ask SEISM to select and test 4 motifs, and only one is discovered with a very low p-value, which is relevant regarding the only known motif from the literature for this transcription factor, which shows that SEISM was able to associate the right number of motifs.
>
> Regarding the other requested changes:
> The partition of the motif space has to be defined without using the data. Otherwise the selective inference procedure will remain uncalibrated, because it will essentially not condition sufficiently. One way to see this is using the procedure with the center of the mesh: when sampling new phenotypes y’ under H0, those falling in the selected mesh will not have their highest association with the center of the mesh, whereas the true phenotype y will.

---

> > ### Author Response · Authors · 2023-04-23
> > **To Reviewer rRMK (2/2)**
> >
> >
> > Regarding the potential correlation between the order of the extracted filter and the  associated p-value, we can provide several answers. First, the designed test procedure is joint: each motif is tested in the context of all other selected filters (equations 16 and 17). By contrast, a sequential testing procedure would test each selected motif in the context of the previously selected ones only, which would indeed make the results dependent on the order of selection. Under the null hypothesis, the calibration for the type 1 error implies the absence of correlation between the order of the filter and the p-value for each motif: regardless of the order, each p-value comes from a uniform distribution between 0 and 1, which is incompatible with the existence of a correlation with the entry order. Under an alternative hypothesis H1, the first selected filters are the ones that capture the signal , while the later filters will be selected on noise. We then expect these first motifs to be associated with a low p-value, while the p-values of the last ones will be uniformly distributed between 0 and 1. Nonetheless, because we use a joint (and not a sequential) testing procedure,  any difference observed between the p-values of the motifs under H1 will only be attributable to the difference between their association strength with the phenotype, not the order in which they were selected.
> > SEISM is designed for biological sequences, because in such a framework the filters of the first layer of a CNN can actually be interpreted very simply using sequence motifs, a widely used tool in the bioinformatics literature. We  are afraid that applying it to another type of data would stray too far from the initial topic of this article. We did not mean to claim more generality in terms of data, and would gladly tune down some parts of the manuscripts if the reviewer believes that they are unsubstantiated. We do claim some potential for generalizing to other architectures beyond one-layer CNNs, and added clarification on this point in our Discussion section.

---

### Review · Reviewer_XYhN · 2023-04-09

**Summary Of Contributions:**

The paper formalizes an approach to interpretability for sequence motif identification for neural networks. The paper does not seem to add any significant methodological advances over prior methods.

**Audience:**

Yes

**Claims And Evidence:**

Yes

**Requested Changes:**

Actually showing an extension that the proposed formalism enables would go a long way to showing a broader audience the utility of the method.

**Strengths And Weaknesses:**

This paper focuses on the formalism of gradient-based interpretability for identifying DNA sequence motifs. Given this topic I don't think I am the right reviewer for analysing the strength of this paper.

As far as I can tell, the paper does not claim to significantly advance the performance over current state of the art, by any metric.

Instead, the argument is that the formalism is defined and would be useful in extending the method further. It is difficult for me to evaluate this analysis. The paper does not articulate explicitly what is possible with the new formalism or show actual results on extending it.

As a result, the reader is left to evaluate the utility of the formalism on its own. However, I'm afraid this is beyond my area of expertise to review. I am not an expert in neural network interpretability or on methods for identifying DNA sequence motifs.

---

> ### Author Response · Authors · 2023-04-23
> **To Reviewer XYhN**
>
> We would like to thank reviewer for their work on this review, despite different areas of expertise.
>
> We would like to answer to two main points :
>
>     •  "The paper does not seem to add any significant methodological advances over prior methods"
>
> → To the best of our knowledge, there currently exists no framework to quantify the uncertainty of the association between interpretable features extracted from neural networks and its outcome, and our submission is the first to define one. The invariance results that we give in Propositions 4.2, 4.3 and 4.4 make it possible to do post-selection inference by sampling either from the null distribution (in a data-split setting) or the conditional null (in a selective inference setting).
>
> In addition, even if our selective inference procedure is derived from existing ones, it adds another methodological advance as all existing frameworks only deal with feature selection over a finite set, while SEISM introduces a meshing strategy to deal with a selection over a continuous finite space. Finally, the sampling strategy proposed in this paper is, to our knowledge, the first case of application of a known method to a continuous feature space.
>
>     • "Actually showing an extension that the proposed formalism enables would go a long way to showing a broader audience the utility of the method. "
>
> → Our method could be extended to any selective inference problem, with a continuous feature space, with only limited hypotheses on the association score (i.e. on the test statistics).   However, we believe that the current application is already significant as it deals with a highly active research in bioinformatics, and we are afraid that extending our methods to other fields of application in this paper would reduce its readability.

---

### Author Response · Authors · 2023-04-23
**To all three reviewers**

Reading your reviews made us realize that we were giving a misleading presentation of our work, for historical reasons. We first worked on the selective inference part, before developing propositions 4.2, 4.3 and 4.4, which are actually necessary for both the selective inference and data split strategies. The current presentation incorrectly suggests that the data-split strategy was already a possible method to test features selected by a neural network, and that our only contribution is the selective inference strategy.

If the reviewers agree, we would like to fix this issue and give the following presentation:

- We define a post-selection inference framework for the features selected by the neural network, using either data-split or selective inference (Section 4), each being more appropriate in a given sample size regime: selective inference is more powerful but less scalable than data-split, making it more appropriate when fewer samples are available.

- Both strategies require sampling under a normal null hypothesis which is composite—several mean vectors define the same null—and depends on unknown parameters. We provide invariance results suggesting a practical procedure that works around these issues (Section 4.6). Without those results, the user would have to make additional assumptions on the data, which may cause the whole procedure to be decalibrated.

The other contributions (formulation of the CNN training as a motif selection step, novel selective inference procedure acting on the selected motifs and software implementing both the data split and selective inference strategies) are unchanged.

This new presentation only entails limited changes in the introduction and in the following sections (sections 1, 2 and 4.6, highlighted in the updated manuscript) and we believe it provides a clearer picture of our contribution. We are grateful to the reviewers for helping us perceive this issue.

---

### Decision · Action_Editors · 2023-06-09

**Recommendation:** Accept as is

**Comment:**

The paper has gone through a thorough review, discussion and the authors have already updated the paper accordingly. The paper has some issues about general applicability and scalability but is valid and therefore meets the TMLR evaluation criteria.

The authors are encouraged to add a more realistic test case for the final version.

Minor comment:

"Over the past decade, neural networks have been successful at making predictions from biological sequences, especially in the context of regulatory genomics." Arguably, there has been more activity in the protein space. So suggest to remove ", especially in the context of regulatory genomics".

**Audience:**

Yes

**Claims And Evidence:**

Yes